# Structural Pruning for Diffusion Models

**Gongfan Fang    Xinyin Ma    Xinchao Wang**[*]
National University of Singapore
gongfan@u.nus.edu, maxinyin@u.nus.edu, xinchao@nus.edu.sg

## Abstract

Generative modeling has recently undergone remarkable advancements, primarily propelled by the transformative implications of Diffusion Probabilistic Models (DPMs). The impressive capability of these models, however, often entails significant computational overhead during both training and inference. To tackle this challenge, we present *Diff-Pruning*, an efficient compression method tailored for learning lightweight diffusion models from pre-existing ones, without the need for extensive re-training. The essence of Diff-Pruning is encapsulated in a Taylor expansion over *pruned timesteps*, a process that disregards non-contributory diffusion steps and ensembles informative gradients to identify important weights. Our empirical assessment, undertaken across several datasets highlights two primary benefits of our proposed method: 1) *Efficiency:* it enables approximately a 50% reduction in FLOPs at a mere 10% to 20% of the original training expenditure; 2) *Consistency*: the pruned diffusion models inherently preserve generative behavior congruent with their pre-trained models. Code is available at `https://github.com/VainF/Diff-Pruning`.

## 1   Introduction

Generative modeling has undergone significant advancements in the past few years, largely propelled by the advent of Diffusion Probabilistic Models (DPMs) [18, 41, 37]. These models have derived numerous applications ranging from text-to-image generation [40], image editing [58], image translation[45], and even discriminative tasks [2, 1]. The incredible power of DPMs, however, often comes at the expense of considerable computational overhead during both training [49] and inference [43]. This trade-off between performance and efficiency presents a critical challenge in the broader application of these models, particularly in resource-constrained environments.

In the literature, huge efforts have been made to improve diffusion models, which primarily revolved around three broad themes: improving model architectures [41, 39, 52], optimizing training methods [49, 11] and accelerating sampling [46, 43, 12]. As a result, a multitude of well-trained diffusion models has been created in these valuable works, showcasing their potential for various applications [48]. However, the notable challenge still remains: the absence of a general compression method that enables the efficient reuse and customization of these pre-existing models without heavy re-training. Overcoming this gap is of paramount importance to fully harness the power of pre-trained diffusion models and facilitate their widespread application across different domains and tasks.

In this work, we demonstrate the remarkable effectiveness of structural pruning  [23, 8, 26, 4] as a method for compressing diffusion models, which offers a flexible trade-off between efficiency and quality. Structural pruning is a classic technique that effectively reduces model sizes by eliminating redundant parameters and sub-structures from networks. While it has been extensively studied in discriminative tasks such as classification [16], detection [54], and segmentation [13], applying structural pruning techniques to Diffusion Probabilistic Models poses unique challenges that necessitate

---

[*]Corresponding author

37th Conference on Neural Information Processing Systems (NeurIPS 2023).

a rethinking of traditional pruning strategies. For example, the iterative nature of the generative process in DPMs, the models' sensitivity to small perturbations in different timesteps, and the intricate interplay in the diffusion process collectively create a landscape where conventional pruning strategies often fall short.

To this end, we introduce a novel approach called *Diff-Pruning*, explicitly tailored for the compression of diffusion models. Our method is motivated by the observation in previous works [41, 52] that different stages in the diffusion process contribute variably to the generated samples. At the heart of our method lies a Taylor expansion over pruned timesteps, which deftly balances the image content, details, and the negative impact of noisy diffusion steps during pruning. Initially, we show that the objective of diffusion models at late timesteps ($t \to T$) prioritize the high-level content of the generated images during pruning, while the early ones ($t \to 0$) refine the images with finer details. However, it is also observed that, when using Taylor expansion for pruning, the noisy stages with large $t$ can not provide informative gradients for importance estimation and can even harm the compressed performance. Therefore, we propose to model the trade-off between contents, details, and noises as a pruning problem of the diffusion timesteps, which leads to an efficient and flexible pruning algorithm for diffusion models.

Through extensive empirical evaluations across diverse datasets, we demonstrate that our method achieves substantial compression rates while preserving and in some cases even improving the generative quality of the models. Our experiments also highlight two significant features of Diff-Pruning: efficiency and consistency. For example, when applying our method to an off-the-shelf diffusion model pre-trained on LSUN Church [57], we achieve an impressive compression rate of 50% FLOPs, with only 10% of the training cost required by the original models, equating to 0.5 million steps compared to the 4.4 million steps of the pre-existing models. Furthermore, we have thoroughly assessed the generative behavior of the compressed models both qualitatively and quantitatively. Our evaluations demonstrate that the compressed model can effectively preserve a similar generation behavior as the pre-trained model, meaning that when provided with the same inputs, both models yield consistent outputs. Such consistency further reveals the practicality and reliability of Diff-Pruning as a compression method for diffusion models.

In summary, this paper introduces Diff-Pruning as an efficient method for compressing Diffusion Probabilistic Models, which is able to achieve compression with only 10% to 20% of the training costs compared to pre-training. This work may serve as an initial baseline and provide a foundation for future research aiming to enhance the quality and consistency of compressed diffusion models.

## 2   Ralted Works

**Efficient Diffusion Models**   The existing methodologies principally address the efficiency issues associated with diffusion models via three primary strategies: the refinement of network architectures [41, 52, 37], the enhancement of training procedures [11, 49], and the acceleration of sampling [18, 27, 12]. Diffusion models typically employ U-Net models as denoisers, of which the efficiency can be improved via the introduction of hierarchical designs [40] or by executing the training within a novel latent space [41, 19, 25]. Recent studies also suggest integrating more efficient layers or structures into the denoiser to bolster the performance of the U-Net model [52, 39], thereby facilitating superior image quality learning during the training phase. Moreover, a considerable number of studies concentrate on amplifying the training efficiency of diffusion models, with some demonstrating that the diffusion training can be expedited by modulating the weights allocated to distinct timesteps [43, 11]. The training efficiency can also be advanced by learning diffusion models at the patch level [49]. In addition, some approaches underscore the efficiency of sampling, which typically does not necessitate the retraining of diffusion models [27]. In this area, numerous studies aim to diminish the required steps through methods such as early stopping [34] or distillation [43].

**Network Pruning**   In recent years, the field of network acceleration [59, 3, 20, 53, 51, 29, 30] has seen notable progress through the deployment of network pruning techniques [31, 16, 33, 23, 14, 5, 15]. The taxonomy of pruning methodologies typically bifurcates into two main categories: structural pruning [23, 6, 56, 26, 56] and unstructured pruning [38, 7, 44, 22]. The distinguishing trait of structural pruning is its ability to physically eliminate parameters and substructures from networks, while unstructured pruning essentially masks parameters by zeroing them out [8, 4]. However, the preponderance of network pruning research is primarily focused on discriminative

tasks, particularly classification tasks [16]. A limited number of studies have ventured into examining the effectiveness of pruning in generative tasks, such as GAN compression [24, 47]. Moreover, the application of structural pruning techniques to Diffusion Probabilistic Models introduces unique challenges that demand a reevaluation of conventional pruning strategies. In this work, we introduce the first dedicated method explicitly designed for pruning diffusion models, which may serve as a useful baseline for future works.

## 3 Diffusion Model Objectives

Given a data distribution $q(\boldsymbol{x})$, diffusion models aim to model a generative distribution $p_\theta(\boldsymbol{x})$ to approximate $q(\boldsymbol{x})$, taking the form

$$p_\theta(\boldsymbol{x}) = \int p_\theta(\boldsymbol{x}_{0:T}) d\boldsymbol{x}_{1:T}, \qquad \text{where} \quad p_\theta(\boldsymbol{x}_{0:T}) := p(\boldsymbol{x}_T) \prod_{t=1}^{T} p_\theta(\boldsymbol{x}_{t-1}|\boldsymbol{x}_t) \tag{1}$$

And $\boldsymbol{x}_1, ..., \boldsymbol{x}_T$ refer to the latent variables, which contribute to the joint distribution $p_\theta(\boldsymbol{x}_{0:T})$ with learned Gaussian transitions $p_\theta(\boldsymbol{x}_{t-1}|\boldsymbol{x}_t) = \mathcal{N}(\boldsymbol{x}_{t-1}; \mu_\theta(\boldsymbol{x}_t, t), \Sigma_\theta(\boldsymbol{x}_t, t))$. Diffusion Models involve two opposite processes: a forward (diffusion) process $q(\boldsymbol{x}_t|\boldsymbol{x}_{t-1}) = \mathcal{N}(\boldsymbol{x}_t; \sqrt{1-\beta_t}\boldsymbol{x}_{t-1}, \beta_t I)$ that adds noises to the $\boldsymbol{x}_{t-1}$, based on a pre-defined variance schedule $\beta_{1:T}$; and a reverse process $q(\boldsymbol{x}_{t-1}|\boldsymbol{x}_t)$ which "denoises" the observation $\boldsymbol{x}_t$ to get $\boldsymbol{x}_{t-1}$. Using the notation $\alpha_t = 1 - \beta_t$ and $\bar{\alpha}_t = \prod_{s=1}^{t} \alpha_s$, DDPMs [18] trains a noise predictor with the objective:

$$\mathcal{L}(\boldsymbol{\theta}) := \mathbb{E}_{t, \boldsymbol{x}_0 \sim q(\boldsymbol{x}), \boldsymbol{\epsilon} \sim \mathcal{N}(0,1)} \left[ \| \boldsymbol{\epsilon} - \boldsymbol{\epsilon}_{\boldsymbol{\theta}}(\sqrt{\bar{\alpha}_t}\boldsymbol{x}_0 + \sqrt{1-\bar{\alpha}_t}\boldsymbol{\epsilon}, t) \|^2 \right] \tag{2}$$

where $\boldsymbol{\epsilon}$ is a random noise drawn from a fixed Gaussian distribution and $\boldsymbol{\epsilon}_\theta$ refers to a learned noise predictor, which is usually an U-Net autoencoder [42] in practice. After training, synthetic images $\boldsymbol{x}_0$ can be sampled through an iterative process from a noise $\boldsymbol{x}_T \sim \mathcal{N}(\boldsymbol{0}, \boldsymbol{1})$ with the formular:

$$\boldsymbol{x}_{t-1} = \frac{1}{\sqrt{\alpha_t}} \left( \boldsymbol{x}_t - \frac{\beta_t}{\sqrt{1-\bar{\alpha}_t}} \boldsymbol{\epsilon}_{\boldsymbol{\theta}}(\boldsymbol{x}_t, t) \right) + \sigma_t \boldsymbol{z} \tag{3}$$

where $\boldsymbol{z} \sim \mathcal{N}(\boldsymbol{0}, \boldsymbol{I})$ for steps $t > 1$ and $\boldsymbol{z} = \boldsymbol{0}$ for $t = 1$. In this work, we aim to craft a lightweight $\boldsymbol{\epsilon}_{\boldsymbol{\theta}'}$ by removing redundant parameters of $\boldsymbol{\epsilon}_{\boldsymbol{\theta}}$, which are expected to produce similar $\boldsymbol{x}_0$ while the same $\boldsymbol{x}_T$ are presented.

## 4 Structrual Pruning for Diffusion Models

Given the parameter $\boldsymbol{\theta}$ of a pre-trained diffusion model, our goal is to craft a lightweight $\boldsymbol{\theta}'$ by removing sub-structures from the network following existing paradigms [35, 8]. Without loss of generality, we assume that the parameter $\boldsymbol{\theta}$ is a simple 2-D matrix, where each sub-structure $\boldsymbol{\theta}_i = [\theta_{i0}, \theta_{i1}, ..., \theta_{iK}]$ is a row vector that contains $K$ scalar parameters. Structural pruning aims to find a sparse parameter matrix $\boldsymbol{\theta}'$ that maximally preserves the original performance. Thus, a natural choice is to optimize the loss disruption caused by pruning:

$$\min_{\boldsymbol{\theta}'} |\mathcal{L}(\boldsymbol{\theta}') - \mathcal{L}(\boldsymbol{\theta})|, \qquad \text{s.t. } \|\boldsymbol{\theta}'\|_0 \leq s \tag{4}$$

The term $|\boldsymbol{\theta}'|_0$ denotes the L-0 norm of the parameters, which counts the number of non-zero row vectors, and $s$ represents the sparsity of the pruned model. Nevertheless, due to the iterative nature intrinsic to diffusion models, the training objective, denoted by $\mathcal{L}$, can be perceived as a composition of $T$ interconnected tasks: $\{\mathcal{L}_1, \mathcal{L}_2, ..., \mathcal{L}_T\}$. Each task affects and depends on the others, thereby posing a new challenge distinct from traditional pruning problems, which primarily concentrate on optimizing a single objective. In light of the pruning objective as defined in Equation 4, we initially delve into the individual contributions of each loss component, $\mathcal{L}_t$ in pruning, and subsequently propose a tailored method, Diff-Pruning, designed for diffusion models pruning.

**Taylor Expansion at $\mathcal{L}_t$** Initially, we need to model the contribution of $\mathcal{L}_t$ for structural pruning. This work leverages Taylor expansion [35] on $\mathcal{L}_t$ to linearly approximate the loss disruption:

$$\mathcal{L}_t(\boldsymbol{\theta}') = \mathcal{L}_t(\boldsymbol{\theta}) + \nabla \mathcal{L}_t(\boldsymbol{\theta})(\boldsymbol{\theta}' - \boldsymbol{\theta}) + O(\|\boldsymbol{\theta}' - \boldsymbol{\theta}\|^2)$$

$$\Rightarrow \mathcal{L}_t(\boldsymbol{\theta}') - \mathcal{L}_t(\boldsymbol{\theta}) = \nabla \mathcal{L}_t(\boldsymbol{\theta})(\boldsymbol{\theta}' - \boldsymbol{\theta}) + O(\|\boldsymbol{\theta}' - \boldsymbol{\theta}\|^2) \tag{5}$$

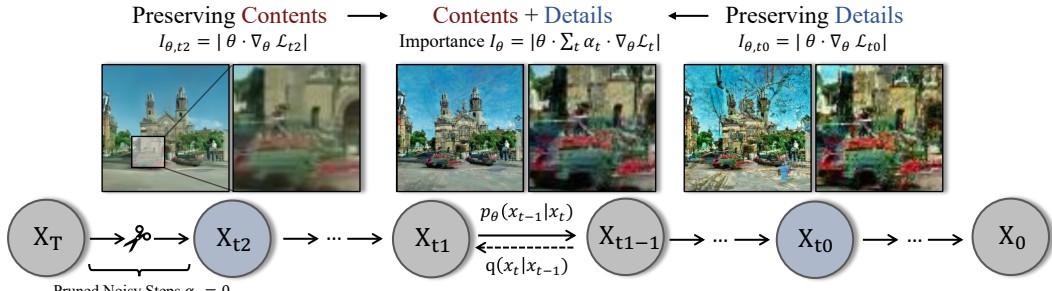

Figure 1: Diff-Pruning leverages Taylor expansion at pruned timesteps to estimate the importance of weights, where early steps focus on local details like edges and color and later ones pay more attention to contents such as object and shape. We propose a simple thresholding method to trade off these factors with a binary weight $\alpha_t \in \{0, 1\}$, leading to a practical algorithm for diffusion models. The generated images produced by 5%-pruned DDPMs (without post-training) are illustrated.

Taylor expansion offers a robust framework for network pruning, as it can estimate the loss disruption using first-order gradients. To evaluate the importance of an individual weight $\boldsymbol{\theta}_{ik}$, we can simply set $\boldsymbol{\theta'}_{ik} = 0$ in Equation 5, which results in the following importance criterion:

$$
\begin{aligned}
\mathcal{I}_t(\boldsymbol{\theta}_{ik}, \boldsymbol{x}) &= |\mathcal{L}_t(\boldsymbol{\theta}|_{\boldsymbol{\theta}_{ik}=0}) - \mathcal{L}_t(\boldsymbol{\theta})| \\
&= |(\boldsymbol{\theta}_{i0} - \boldsymbol{\theta}_{i0}) \cdot \nabla_{\boldsymbol{\theta}_{i0}} + \cdots + (0 - \boldsymbol{\theta}_{ik}) \cdot \nabla_{\boldsymbol{\theta}_{ik}} + \cdots + (\boldsymbol{\theta}_{iK} - \boldsymbol{\theta}_{iK}) \cdot \nabla_{\boldsymbol{\theta}_{iK}}| \quad (6) \\
&= |\boldsymbol{\theta}_{ik} \cdot \nabla_{\boldsymbol{\theta}_{ik}} \mathcal{L}_t(\boldsymbol{\theta}, \boldsymbol{x})|
\end{aligned}
$$

where $\nabla_{\boldsymbol{\theta}_{ik}}$ refer to $\nabla_{\boldsymbol{\theta}_{ik}} \mathcal{L}_t(\boldsymbol{\theta}, \boldsymbol{x})$. In structural pruning, we aim to remove the entire vector $\boldsymbol{\theta'}_i$ concurrently. The standard Taylor expansion for multiple variables, as described in the literature [9], advocates using $|\sum_k \boldsymbol{\theta}_{ik} \cdot \nabla_{\boldsymbol{\theta}_{ik}} \mathcal{L}_t(\boldsymbol{\theta}, \boldsymbol{x})|$ for importance estimation. This method exclusively takes into account the loss difference between the initial state $\boldsymbol{\theta}$ and the final states $\boldsymbol{\theta'}$. However, considering the iterative nature of diffusion models, even minor fluctuations in loss can influence the final generation results. To this end, we propose to aggregate the influence of removing each parameter as the final importance. This modification models cumulative loss disturbance induced by each $\boldsymbol{\theta}_{ik}$'s removal and leads to a slightly different score function for structural pruning:

$$
\mathcal{I}_t(\boldsymbol{\theta}_i, \boldsymbol{x}) = \sum_k |\mathcal{L}_t(\boldsymbol{\theta}|_{\boldsymbol{\theta}_{ik}=\mathbf{0}}) - \mathcal{L}_t(\boldsymbol{\theta})| = \sum_k |\boldsymbol{\theta}_{ik} \cdot \nabla_{\boldsymbol{\theta}_{ik}} \mathcal{L}_t(\boldsymbol{\theta}, \boldsymbol{x})| \quad (7)
$$

In the following sections, we utilize Equation 7 as the importance function to identify non-critical parameters in diffusion models.

**The Contribution of $\mathcal{L}_t$.** With the Taylor expansion framework, we further explore the contribution of different loss terms $\{\mathcal{L}_1, ..., \mathcal{L}_T\}$ during pruning. We consider the functional error $\boldsymbol{\delta}_t = \boldsymbol{\epsilon}_{\boldsymbol{\theta'}}(\boldsymbol{x}, t) - \boldsymbol{\epsilon}_{\boldsymbol{\theta}}(\boldsymbol{x}, t)$ which represents the prediction error for the same inputs at time step $t$. The reverse process allows us to exam the effects $\boldsymbol{\delta}_{t\to 0}$ on the generated images $x_0$ by iteratively applying the Equation 3 starting from $\boldsymbol{\epsilon}_{\boldsymbol{\theta'}}(\boldsymbol{x}, t) = \boldsymbol{\epsilon}_{\boldsymbol{\theta}}(\boldsymbol{x}, t) + \boldsymbol{\delta}_t$. At the $t - 1$ step, it leads to the error $\boldsymbol{\delta}_{t-1}$ derived as:

$$
\begin{aligned}
\delta_{t-1} &= \left[ \frac{1}{\sqrt{\alpha_t}} \left( \boldsymbol{x}_t - \frac{\beta_t}{\sqrt{1 - \bar{\alpha}_t}} \boldsymbol{\epsilon}_{\boldsymbol{\theta}}(\boldsymbol{x}_t, t) \right) + \sigma_t \boldsymbol{z} \right] - \left[ \frac{1}{\sqrt{\alpha_t}} \left( \boldsymbol{x}_t - \frac{\beta_t}{\sqrt{1 - \bar{\alpha}_t}} (\boldsymbol{\epsilon}_{\boldsymbol{\theta}}(\boldsymbol{x}_t, t) + \delta_t) \right) + \sigma_t \boldsymbol{z} \right] \\
&= \frac{1}{\sqrt{\alpha_t}} \frac{\beta_t}{\sqrt{1\bar{\alpha}_t}} \delta_t
\end{aligned}
$$
$$(8)$$

This error has a direct impact on the subsequent input, given by $x'_{t-1} = x_{t-1} + \delta_{t-1}$. By checking Equation 3, we can observe that these perturbed inputs can further trigger a chained effect through both $\frac{1}{\sqrt{\alpha_{t-1}}} x'_{t-1}$ and $-\frac{1}{\sqrt{\alpha_{t-1}}} \frac{\beta_{t-1}}{\sqrt{1-\bar{\alpha}_{t-1}}} \boldsymbol{\epsilon}_{\boldsymbol{\theta'}}(\boldsymbol{x}'_{t-1}, t-1)$. In the first term, the distortion progressively amplifies by a factor $\frac{1}{\sqrt{\alpha_{t-1}}} > 1$, which means that this error will be enhanced throughout the generation process. Regarding the second term, pruning affects both the functionality parameterized by $\boldsymbol{\theta'}$ and the inputs $\boldsymbol{x}'_{t-1}$, which contributes to the final results in a nonlinear and more complicated manner, resulting in a more substantial disturbance on the generated images.

---

**Algorithm 1** Diff-Pruning

---

**Input:** A pretrained diffusion model $\boldsymbol{\theta}$, a dataset $X$, a threshold $\mathcal{T}$ and a pruning ratio $p\%$
**Output:** The pruned diffusion model $\boldsymbol{\theta}'$

1: $\mathcal{L}_{max} = \mathbf{0}$
2: $x = \text{mini-batch}(X)$;
3: $\boldsymbol{\epsilon} \sim \mathcal{N}(0,1)$
4: ▷ Accumulating gradients over partial steps with the threshold $\mathcal{T}$
5: **for** $t$ in $[0,1,2,...,T]$ **do**:
6:    $\mathcal{L}_t = \|\boldsymbol{\epsilon} - \boldsymbol{\epsilon_\theta}(\sqrt{\bar{\alpha}_t}\boldsymbol{x} + \sqrt{1-\bar{\alpha}_t}\boldsymbol{\epsilon}, t)\|^2$;                 ▷ Equation 2
7:    $\mathcal{L}_{max} = \max(\mathcal{L}_{max}, \mathcal{L}_t)$
8:    **if** $\mathcal{L}_t/\mathcal{L}_{max} \leq \mathcal{T}$ **then**
9:       **break**;                             ▷ The threshold in Equation 10
10:    **end if**
11:    $\nabla_{\boldsymbol{\theta}_{ik}}\mathcal{L}_t(\boldsymbol{\theta}, x) = \text{back-propagation}(\mathcal{L}_t(\boldsymbol{\theta}, x))$
12: **end for**
13: ▷ Estimating the importance of sub-structure $\boldsymbol{\theta}_i$ with the accumulated $t$-step gradients
14: $\mathcal{I}(\boldsymbol{\theta}_i, x) = \sum_k |\boldsymbol{\theta}_{ik} \cdot \sum_{s=0}^t \nabla_{\boldsymbol{\theta}_{ik}}\mathcal{L}_s(\boldsymbol{\theta}, x)|$            ▷ Equation 10
15: ▷ Pruning and finetuning
16: Remove $p\%$ channels in each layer to obtain $\boldsymbol{\theta}'$.
17: Finetune the pruned model $\boldsymbol{\theta}'$ on $X$
18: **return** $\boldsymbol{\theta}'$

---

As a result, prediction errors occurring at larger $t$ tend to have a larger impact on the images due to the chain effect, which might change the global content of generated images. Conversely, smaller $t$ values focus on refining the images with relatively small modifications. These findings align with our empirical examination using Taylor expansion as illustrated in Figure 1, as well as the observation in previous works [18, 52], which shows that diffusion models tend to generate object-level information at larger $t$ values and fine-tune the features at smaller ones. To this end, we model the pruning problem as a weighted trade-off between contents and details by introducing $\alpha_t$, which acts as a weighting variable for different timesteps $t$. Nevertheless, unconstrained reweighting can be highly inefficient, as it entails exploring a large parameter space for $\alpha_t$ and requires at least $T$ forward-backward passes for Taylor expansion. This results in a vast sampling space and can lead to inaccuracies in the linear approximation. To address this issue, we simplify the re-weighting strategy by treating it as a "pruning problem", where $\alpha_t$ takes the value of either 0 or 1 for all steps, allowing us to only leverage partial steps for pruning. The general importance metric is modeled as the following.

$$\mathcal{I}(\boldsymbol{\theta}_i, \mathbf{x}) = \sum_k \left| \boldsymbol{\theta}_{ik} \cdot \sum_t \alpha_t \nabla_{\boldsymbol{\theta}_{ik}}\mathcal{L}_t(\boldsymbol{\theta}, \mathbf{x}) \right|, \qquad \text{s.t. } \alpha_t \in \{0,1\} \tag{9}$$

**Taylor Score over Pruned Timesteps.** In Equation 9, we try to remove some "unimportant" timesteps in the diffusion process so as to enable an efficient and stable approximation for partial steps. Our empirical results, as will be discussed in the experiments, indicate two key findings. Firstly, we note that the timesteps responsible for generating content are not exclusively found towards the end of the diffusion process ($t \to T$). Instead, there are numerous noisy and redundant timesteps that contribute minorly to the overall generation, which is similar to the observations in the related work [34]. Secondly, we discovered that employing the full-step objective can sometimes yield suboptimal results compared to using a partial objective. We attribute this negative impact to the presence of converged gradients in the noisy steps ($t \to T$). Taylor approximation in Equation 5 comprises both first-order gradients and higher-order terms. When the loss $\mathcal{L}_t$ converges, the loss curve is predominantly influenced by the higher-order terms rather than the first-order gradients we utilize. Our experiments on several datasets and diffusion models show that the loss term $\mathcal{L}_t$ rapidly approaches 0 as $t \to T$. For example in Figure 5, the relative loss $\frac{\mathcal{L}_t}{\mathcal{L}_{max}}$ of a pre-trained diffusion model for CIFAR-10 decreases to 0.05 when $t = 250$. Consequently, a full Taylor expansion can accumulate a considerable amount of noisy gradients from these converged or unimportant steps, resulting in an inaccurate estimation of weight importance.

Considering the significant impact of larger timesteps, it is necessary to incorporate them for importance estimation. To address this problem, Equation 9 naturally provides a simple and practical thresholding strategy for pruning. To achieve this, we introduce a threshold parameter $\mathcal{T}$ based on the relative loss $\frac{\mathcal{L}_t}{\mathcal{L}_{max}}$. Those timesteps with a relative loss below this threshold, i.e., $\frac{\mathcal{L}_t}{\mathcal{L}_{max}} < \mathcal{T}$, are considered uninformative and are disregarded by setting $\alpha_t = 0$, which yields the finalized importance score:

$$\mathcal{I}(\boldsymbol{\theta}_i, \mathbf{x}) = \sum_k \left| \boldsymbol{\theta}_{ik} \cdot \sum_{\{t | \frac{\mathcal{L}_t}{\mathcal{L}_{max}} > \mathcal{T}\}} \nabla_{\boldsymbol{\theta}_{ik}} \mathcal{L}_t(\boldsymbol{\theta}, \mathbf{x}) \right| \tag{10}$$

In practice, we need to select an appropriately large value for $\mathcal{T}$ to strike a well-balanced preservation of details and content, while also avoiding uninformative gradients from noisy loss terms. The full algorithm is summarized in Alg. 1.

## 5 Experiments

### 5.1 Settings

**Datasets and Models**   The efficacy of Diff-Pruning is empirically validated across six diverse datasets, including CIFAR-10 (32×32)[21], CelebA-HQ (64×64)[32], LSUN Church (256×256), LSUN Bedroom (256×256) [57] and ImageNet-1K (256×256). We focus on two popular DPMs in our experiments, i.e., Denoising Diffusion Probability Models (DDPMs) [18] and Latent Diffusion Models (LDMs) [41]. For the sake of reproducibility, we utilize off-the-shelf DPMs from [18] and [41] as pre-trained models and prune these models in a one-shot fashion[23].

**Evaluation Metrics**   In this paper, we concentrate primarily on three types of metrics: 1) Efficiency metrics, which include the number of parameters (#Params) and Multiply-Add Accumulation (MACs); 2) Quality metric, namely the Frechet Inception Distance (FID) [17]; and 3) Consistency metric, represented by Structural Similarity (SSIM) [50]. Unlike previous generative tasks that lacked reference images, we employ the SSIM index to evaluate the similarity between images generated by pre-trained models and pruned models, given identical noise inputs. We deplpy a 250-step DDIM sampler [46] for ImageNet and a 100-step DDIM sampler for other experiments.

### 5.2 An Simple Benchmark for Diffusion Pruning

**Scratch Training v.s. Pruning.**   Table 1 shows our results on CIFAR-10 and CelebA-HQ. The first baseline method that piques our interest is scratch training. Numerous studies on network pruning [10] suggest that training a compact network from scratch can be a formidable competitor. To ensure a fair comparison, we create randomly initialized networks with the same architecture as the pruned ones for scratch training. Our results reveal that scratch training demands relatively more steps to reach convergence. This suggests that training lightweight models from scratch may not be an efficient and economical approach, given its training cost is comparable to that of pre-trained models. Conversely, we observe that all pruning methods are able to converge within approximately 100K steps and outperform scratch training in terms of FID and SSIM scores. Thus, pruning emerges as a potent technique for compressing pre-trained Diffusion Models.

**Pruning Criteria.**   A significant aspect of network pruning is the formulation of pruning criteria, which serve to identify superfluous parameters within networks. Due to the absence of dedicated work on Diffusion model pruning, we adapted three basic pruning methods from discriminative tasks: random pruning, magnitude-based pruning [16], and Taylor-based pruning [36], which we refer to as Random, Magnitude, and Taylor respectively in subsequent sections. For a given parameter $\boldsymbol{\theta}$, Random assigns importance scores derived from a uniform distribution to each $\boldsymbol{\theta}_i$ randomly, denoted as $\mathcal{I}(\boldsymbol{\theta}) \sim \mathcal{U}(0, 1)$. This results in a straightforward baseline devoid of any prior or bias, and has been shown to be a competitive baseline for pruning [28]. Magnitude subscribes to the "smaller-norm-less-informative" hypothesis [23, 55], modelling the weight importance as $\mathcal{I}(\boldsymbol{\theta}) = |\boldsymbol{\theta}|$. In contrast, Taylor is a data-driven criterion that measures importance as $\mathcal{I}(\boldsymbol{\theta}, x) = |\boldsymbol{\theta} \cdot \nabla_{\boldsymbol{\theta}} \mathcal{L}(x, \boldsymbol{\theta})|$, which aims to minimize loss change as discussed in our method. As shown in 1, an intriguing phenomenon is that these three baseline methods do not maintain a consistent ranking on these two datasets. For

| CIFAR-10 32 × 32 (100 DDIM steps) | | | | | |
|---|---|---|---|---|---|
| **Method** | **#Params ↓** | **MACs ↓** | **FID ↓** | **SSIM ↑** | **Train Steps ↓** |
| Pretrained | 35.7M | 6.1G | 4.19 | 1.000 | 800K |
| Scratch Training | | | 9.88 | 0.887 | 100K |
| Scratch Training | | | 5.68 | 0.905 | 500K |
| Scratch Training | 19.8M | 3.4G | 5.39 | 0.905 | 800K |
| Random Pruning | | | 5.62 | 0.926 | 100K |
| Magnitude Pruning | | | 5.48 | 0.929 | 100K |
| Taylor Pruning | | | 5.56 | 0.928 | 100K |
| Ours ($\mathcal{T} = 0.00$) | | | 5.49 | 0.932 | 100K |
| Ours ($\mathcal{T} = 0.02$) | 19.8M | 3.4G | 5.44 | 0.931 | 100K |
| Ours ($\mathcal{T} = 0.05$) | | | **5.29** | **0.932** | 100K |

| CelebA-HQ 64 × 64 (100 DDIM steps) | | | | | |
|---|---|---|---|---|---|
| **Method** | **#Params** | **MACs** | **FID** | **SSIM** | **Train Steps** |
| Pretrained | 78.7M | 23.9G | 6.48 | 1.000 | 500K |
| Scratch Training | | | 7.08 | 0.833 | 100K |
| Scratch Training | | | 6.73 | 0.867 | 300K |
| Scratch Training | 43.7M | 13.3G | 6.71 | 0.869 | 500K |
| Random Pruning | | | 6.70 | 0.874 | 100K |
| Magnitude Pruning | | | 7.08 | 0.870 | 100K |
| Taylor Pruning | | | 6.64 | 0.880 | 100K |
| Ours ($\mathcal{T} = 0.00$) | | | **6.24** | **0.885** | 100K |
| Ours ($\mathcal{T} = 0.02$) | 43.7M | 13.3G | 6.45 | 0.878 | 100K |
| Ours ($\mathcal{T} = 0.05$) | | | 6.52 | 0.878 | 100K |

Table 1: Diffusion pruning on CIFAR-10 and CelebA. We leverage Frechet Inception Distance (FID) and Structural Similarity (SSIM) to estimate the quality and similarity of generated samples under the same random seed. A larger SSIM score means more consistent generation.

instance, while Magnitude achieves the best FID performance among the three on CIFAR-10, it performs poorly on CelebA datasets. In contrast, our method delivers stable improvements over baseline methods, demonstrating superior performance on both datasets. Remarkably, our method even surpasses the pre-trained model on CelebA-HQ, with only 100K optimizations. Nonetheless, performance degradation is observed on CIFAR-10, which can be attributed to its more complex scene and a larger number of categories.

### 5.3 Pruning at Higher Resolutions

**DDPMs on LSUN**    To further validate the efficiency and effectiveness of our proposed Diff-Pruning, we perform pruning experiments on two 256×256 scene datasets, LSUN Church, and LSUN Bedroom [57]. The pre-trained models from [18] require around 2.4M and 4.4M training steps, which can be quite time-consuming in practice. We demonstrate that Diff-Pruning can compress these pre-existing models using only 10% of the standard training resources. We report the number of parameters, MACs, and FID scores in Table 2, and compare the pruned methods to the pre-trained ones as well as those trained from scratch. Results show that the pruned model converges with a passable FID score in 10% of the standard steps, while a model trained from scratch is still severely under-fitted. Nevertheless, we also discover that compressing a model trained on large-scale datasets, such as LSUN Bedroom, which contains 300K images, proves to be quite challenging with a very limited number of training steps. We show that, in the supplementary materials, the FID scores can be further improved with more training steps. Moreover, we also visualize the generated images in Figure 2 and report the single-image SSIM score to measure the similarity of generated images. By nature, the pruned models can preserve similar generation capabilities since they inherit most parameters from the pre-trained models.

**Conditional LDMs on ImageNet**    Table 3 and Figure 3 illustrate the pruning results of LDM pre-trained on ImageNet-1K. An LDM consists of an encoder, a decoder, and a U-Net model. Around

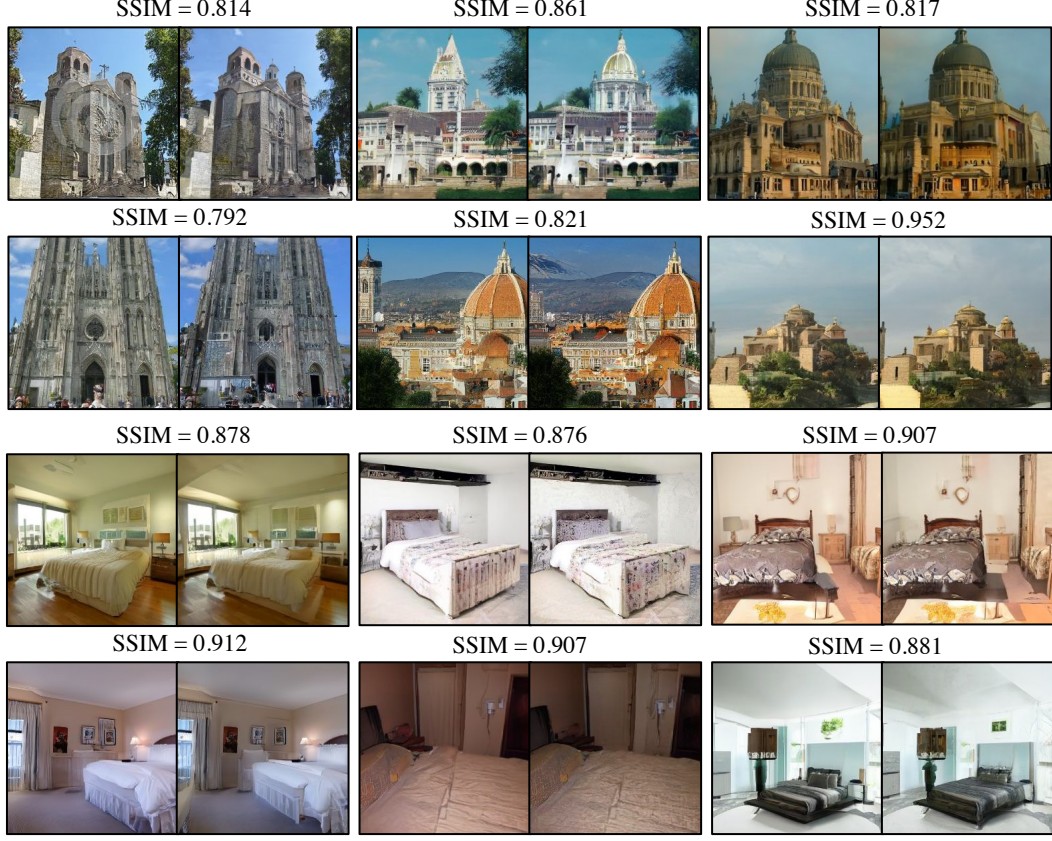

Figure 2: Generated images of the pre-trained models [18] (left) and the pruned models (right) on LSUN Church and LSUN Bedroom. SSIM measures the similarity between generated images.

| LSUN-Church 256 × 256 (DDIM 100 Steps) | | | | | LSUN-Bedroom 256 × 256 (DDIM 100 Steps) | | | | |
|---|---|---|---|---|---|---|---|---|---|
| **Method** | **#Params** | **MACs** | **FID** | **Steps** | **Method** | **#Params** | **MACs** | **FID** | **Steps** |
| Pretrained | 113.7M | 248.7G | 10.6 | 4.4M | Pretrained | 113.7M | 248.7G | 6.9 | 2.4M |
| Scratch Training | 63.2M | 138.8G | 40.2 | 0.5M | Scratch Training | 63.2M | 138.8G | 50.3 | 0.2M |
| Ours ($\mathcal{T} = 0.01$) | 63.2M | 138.8G | **13.9** | 0.5M | Ours ($\mathcal{T} = 0.01$) | 63.2M | 138.8G | **18.6** | 0.2M |

Table 2: Pruning diffusion models on LSUN Church and LSUN Bedroom.

400M parameters come from the U-Net architecture and only 55M from the autoencoder. Therefore, we mainly focus on the pruning of the U-Net model. We used the threshold $\mathcal{T} = 0.1$ to ignore those converged layers and make the pruning process more efficient. With $T = 0.1$, only 534 steps participate in the pruning process. After importance estimation, we apply a pre-defined channel sparsity of 30% to all layers, leading to a lightweight U-Net with 189.43M parameters. Finally, we finetune the pruned model for only 4 epochs with the official training script, with a scaled learning rate of $0.1 \times lr_{\text{base}}$.

## 5.4 Ablation Study

**Pruned Timesteps.** First, we conduct an empirical study evaluating the partial Taylor expansion over pruned timesteps. This approach prioritizes steps with larger gradients and strives to preserve as much content and detail as possible, thereby enabling more accurate and efficient pruning. The impacts of timestep pruning are demonstrated in Figure 5. We seek to prune a pre-trained diffusion model over a range of steps, spanning from 50 to 1000, after which we utilize the SSIM metric to gauge the distortion induced by pruning. In diffusion models, earlier steps ($t \to 0$) usually present larger gradients compared to the later ones ($t \to T$) [41]. This inherently leads to gradients that have reached a convergence when $t$ is large. In the CIFAR-10 dataset, we find that the optimal SSIM score

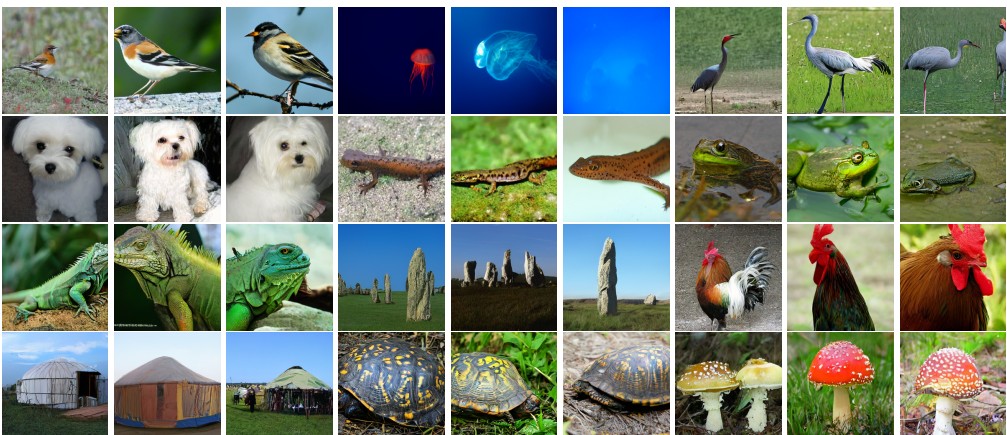

Figure 3: Images sampled from the pruned conditional LDM on ImageNet-1K-256

| Method | #Params ↓ | MACs ↓ | FID ↓ | IS ↑ | Train Steps ↓ |
|---|---|---|---|---|---|
| Pretrained LDM | 400.92M | 99.80G | 3.60 | 247.67 | 2000K |
| Scratch Training | | | 51.45 | 25.69 | 100K |
| Taylor Pruning | 189.43M | 52.71G | 11.18 | 138.97 | 100K |
| Ours ($\mathcal{T} = 0.1$) | | | 9.16 | 201.81 | 100K |

Table 3: Compressing conditional Latent Diffusion Models on ImageNet-1K ($256 \times 256$)

| Pruning Ratios | | | | | | Thresholding | | | |
|---|---|---|---|---|---|---|---|---|---|
| Ratio | #Params | MACs | FID ↓ | SSIM ↑ | | Threshold | Steps | FID ↓ | SSIM ↑ |
| 0% | 35.7M | 6.1G | 4.19 | 1.000 | | $\mathcal{T} = 0.00$ | 1000 | 5.49 | 0.932 |
| 16% | 27.5M | 5.1G | 4.62 | 0.942 | | $\mathcal{T} = 0.01$ | 707 | 5.41 | 0.932 |
| 44% | 19.8M | 3.4G | 5.29 | 0.932 | | $\mathcal{T} = 0.02$ | 433 | 5.44 | 0.931 |
| 56% | 14.3M | 2.7G | 6.36 | 0.922 | | $\mathcal{T} = 0.05$ | 244 | **5.29** | **0.932** |
| 70% | 8.6M | 1.5G | 9.33 | 0.909 | | $\mathcal{T} = 0.10$ | 127 | 5.31 | 0.931 |

Table 4: Pruning with different ratios                Table 5: Pruning with different threshold $\mathcal{T}$

can be attained at around 250 steps, and adding more steps can slightly deteriorate the quality of the synthetic images. This primarily stems from the inaccuracy of the first-order Taylor expansion at converged points, where the gradient no longer provides useful information and can even distort informative gradients through accumulation. However, we observe that the situation differs slightly with the CelebA dataset, where more steps can be beneficial for importance estimation.

**Pruning Ratios.**  Table 4 presents the #Params, MACs, FID, and SSIM scores of models subjected to various pruning ratios based on MACs. Notably, our findings reveal that, unlike CNNs employed in discriminative models, diffusion models exhibit a significant sensitivity to changes in model size. Even a modest pruning ratio of $16\%$ leads to a noticeable degradation in FID score ($4.19 \rightarrow 4.62$). In classification tasks, a perturbation in loss does not necessarily impact the final accuracy; it may only undermine prediction confidence while leaving classification accuracy unaffected. However, in generative models, the FID score is very sensitive, making it more susceptible to domain shift.

**Thresholding.**  In addition, we conducted experiments to investigate the impact of the thresholding parameter $\mathcal{T}$. Setting $\mathcal{T} = 0$ corresponds to a full Taylor expansion at all steps, while $\mathcal{T} > 0$ denotes pruning of certain timesteps during importance estimation. The quantitative findings presented in Table 5 align with the SSIM results depicted in Figure 5. Notably, Diff-Pruning attains optimal performance when the quality of generated images reaches its peak. For datasets such as CIFAR-10, we observed that a 200-step Taylor expansion is sufficient to achieve satisfactory results. Besides, using a full Taylor expansion, in this case, can be detrimental, as it accumulates noisy gradients over approximately 700 steps, which obscures the correct gradient information from earlier steps.

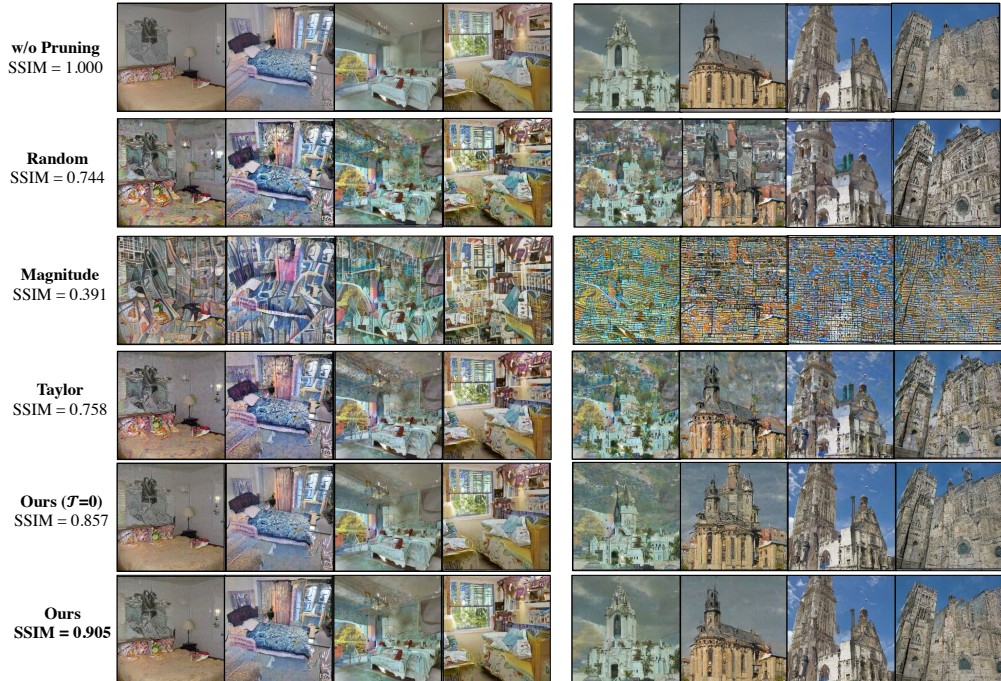

Figure 4: Generated images of 5%-pruned models using different important criteria. We report the SSIM of batched images without post-training.

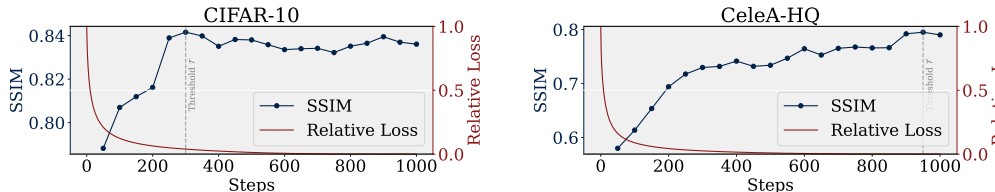

Figure 5: The SSIM of models pruned with different numbers of timesteps. For CIFAR-10, most of the late timesteps can be pruned safely. For CelebA-HQ, using more steps is consistently beneficial.

**Visualization of Different Importance Criteria.** Figure 4 visualizes the images generated by pruned models using different pruning criteria, including the proposed method with $\mathcal{T} = 0$ (w/o timestep pruning) and $\mathcal{T} > 0$. The SSIM scores of the generated samples are reported for a quantitative comparison. The Diff-Pruning method with $\mathcal{T} > 0$ achieves superior visual quality, with an SSIM score of 0.905 after pruning. It is observed that employing more timesteps in our method could have a negative impact, leading to greater distortion in both textures and contents.

## 6 Conclusion

This work introduces Diff-Pruning, a dedicated method for compressing diffusion models. It utilizes Taylor expansion over pruned timesteps to identify and remove non-critical parameters. The proposed approach is capable of crafting lightweight yet consistent models from pre-trained ones, incurring only about 10% to 20% of the cost compared to pre-training. This work may set an initial baseline for future research that aims at improving both the generation quality and the consistency of pruned diffusion models.

## Acknowledgment

This research is supported by the Ministry of Education, Singapore, under its Academic Research Fund Tier 2 (Award Number: MOE-T2EP20122-0006).

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
