# Structural Pruning for Diffusion Models
## — *Supplementary Materials* —

**Gongfan Fang**   **Xinyin Ma**   **Xinchao Wang**[*]
National University of Singapore
gongfan@u.nus.edu, maxinyin@u.nus.edu, xinchao@nus.edu.sg

In this document, we provide supplementary materials that we cannot fit into the main manuscript due to the page limit. It includes detailed explanations, visualization results, and several quantitative experiments.

## 1   Sub-groups in U-Net Denoiser

This section provides further insights into the coupled structures present in U-Net, which function as denoisers in diffusion models. In the context of structural pruning, it is crucial to prune layers with interdependencies simultaneously to avoid any potential structural issues [3]. To address these dependencies within U-Net, we leverage the use of DepGraph [1], which effectively handles most of the interdependencies. However, we encountered new challenges in the pruning process when GroupNorm [4] was introduced. GroupNorm divides the feature maps into $N$ groups, enforcing the constraint that all groups must have the same size. Consequently, this gives rise to $N$ independent pruning problems, each operating independently of the others. To tackle this, we perform pruning on these $N$ groups separately, thereby facilitating the pruning of denoisers.

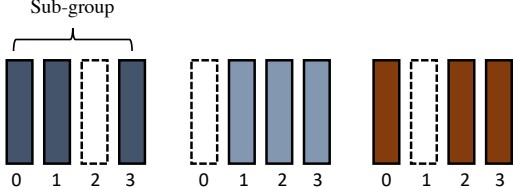

Figure 1: GroupNorm [4] imposes a structural constraint where all groups must be of the same size. As a result, this leads to $N$ independent sub-groups, which must be pruned simultaneously. Different groups are visually highlighted by different colors.

## 2   The Accumulative Loss Disturbance

Taylor expansion approximates the loss function $\mathcal{L}(\boldsymbol{\theta})$ as a linear function of $\boldsymbol{\theta}$ when first-order gradients are used. In structural pruning, a vector $\boldsymbol{\theta}_i$ that contains several parameters will be removed. We discussed two slightly different importance criteria: the standard Taylor expansion for multiple variables,

$$\mathcal{I}_t(\boldsymbol{\theta}_{ik}, \boldsymbol{x}) = |\sum_k \boldsymbol{\theta}_{ik} \cdot \nabla_{\boldsymbol{\theta}_{ik}} \mathcal{L}_t(\boldsymbol{\theta}, \boldsymbol{x})| \tag{1}$$

and the accumulative variant:

$$\mathcal{I}_t(\boldsymbol{\theta}_{ik}, \boldsymbol{x}) = \sum_k |\boldsymbol{\theta}_{ik} \cdot \nabla_{\boldsymbol{\theta}_{ik}} \mathcal{L}_t(\boldsymbol{\theta}, \boldsymbol{x})| \tag{2}$$

---

[*]Corresponding author

37th Conference on Neural Information Processing Systems (NeurIPS 2023).

| CIFAR-10 $32 \times 32$ (100 DDIM steps) | | | | | |
|---|---|---|---|---|---|
| **Method** | **#Params** ↓ | **MACs** ↓ | **FID** ↓ | **SSIM** ↑ | **Train Steps** ↓ |
| Pretrained | 35.7M | 6.1G | 4.19 | 1.000 | 800K |
| Scratch Training | | | 9.88 | 0.887 | 100K |
| Scratch Training | 19.8M | 3.4G | 5.68 | 0.905 | 500K |
| Scratch Training | | | 5.39 | 0.905 | 800K |
| Ours-100K | | | 5.29 | 0.932 | 100K |
| Ours-300K | 19.8M | 3.4G | 5.13 | 0.930 | 300K |
| Ours-500K | | | 5.12 | 0.931 | 500K |
| Ours-100K + KD | | | **5.09** | **0.939** | 100K |

Table 1: Finetuning pruned models with more training steps. The finetuning performance can be boosted with knowledge distillation, which not only improves the FID score but also makes the generated images more consistent.

| LSUN-Bedroom $256 \times 256$ (DDIM 100 Steps) | | | | |
|---|---|---|---|---|
| Method | **#Params** | **MACs** | **FID** | **Steps** |
| Pretrained | 113.7M | 248.7G | 6.9 | 2.4M |
| Scratch Training | 46.5M | 100.7G | 50.3 | 0.2M |
| Ours-0.2M | 46.5M | 100.7G | 18.6 | 0.2M |
| Ours-0.8M | 46.5M | 100.7G | **17.9** | 0.8M |

Table 2: Diffusion models pruning on LSUN Bedroom.

Note that the only difference lies in the position of the summation. It is important to note that Taylor expansion works only for slight changes of $\boldsymbol{\theta'}_i$. Thus, setting a whole vector $\boldsymbol{\theta'}_i$ that contains several parameters to zero can cause significant violates this requirement. Furthermore, Equation 3 will accidentally produce zero disturbance when

$$\sum_k \boldsymbol{\theta}_{ik} \cdot \nabla_{\boldsymbol{\theta}_{ik}} \mathcal{L}_t(\boldsymbol{\theta}, \boldsymbol{x}) = 0, \text{ and } \boldsymbol{\theta}_{ik} \cdot \nabla_{\boldsymbol{\theta}_{ik}} \mathcal{L}_t(\boldsymbol{\theta}, \boldsymbol{x}) \neq 0 \tag{3}$$

even though the removal of each parameter will harm the performance. To remedy the above issues, we use Equation 2 to estimate the cumulative loss disturbance caused by removing single parameters. The results of the proposed method with $\mathcal{T} = 0$, as illustrated in the main paper, show that this improved importance (FID=5.49) works better than the standard Taylor expansion (FID=5.56).

## 3 Improving the Performance of Pruned DPMs

**More training steps.** To enhance pruned models, a straightforward approach is to scale up the finetuning process by increasing the number of steps. The results of our experiments are presented in Table 1. It is easy to observe that our model achieves convergence rapidly. Extending the training of the pruned model to 300K steps yields a slight improvement in FID, yet further increasing the number of steps does not yield significant advantages. Similarly, we also increase the training steps on LSUN-Bedroom from 0.2M to 0.8M. The dataset size of LSUN Bedroom is 44.48GB, which is much larger than the 2.36GB church dataset. It is quite challenging to compress diffusion models trained on LSUN-Bedroom due to the limited capacity of pruned models and the huge data size. With more training steps, the FID score reported in Table 2 can be improved from 18.6 to 17.9.

**Knowledge Distillation.** We conducted further investigations to explore the effectiveness of knowledge distillation in enhancing pruning techniques. In this context, we propose a straightforward optimization objective to train pruned models:

$$\mathcal{L}(\boldsymbol{\theta'}) := \mathbb{E}_{t, \boldsymbol{x}_0, \boldsymbol{\epsilon}} \left[ \| \boldsymbol{\epsilon}_{\boldsymbol{\theta'}}(\sqrt{\bar{\alpha}_t} \boldsymbol{x}_0 + \sqrt{1 - \bar{\alpha}_t} \boldsymbol{\epsilon}, t) - \boldsymbol{\epsilon}_{\boldsymbol{\theta}}(\sqrt{\bar{\alpha}_t} \boldsymbol{x}_0 + \sqrt{1 - \bar{\alpha}_t} \boldsymbol{\epsilon}, t) \|^2 \right] \tag{4}$$

Where $\theta_p$ represents the pruned parameters, our optimization objective aims to align the predictions of the pruned models with those of the pre-trained models. The effectiveness of this objective is demonstrated in Table 1, where it is observed that knowledge distillation significantly enhances the quality and consistency of the generated images.

# 4 Speed Up

Table 3 profiles the pre-trained and the pruned models on a single A5000, with a batch size of 1. We repeat the experiments 50 times and report the average results.

| Method | #Params ↓ | MACs ↓ | Inference Mem ↓ | Train Mem. ↓ | Inference FPS ↑ | Train FPS ↑ |
|---|---|---|---|---|---|---|
| Pretrained LDM | 400.92M | 99.80G | 11.03GB | 14.64GB | 12.87 | 4.26 |
| Pruned LDM | 189.43M | 52.71G | 9.58GB | 11.95GB | 19.83 | 6.37 |
| Pretrained DDPM | 113.7M | 248.7G | 3.35GB | 5.59GB | 28.66 | 9.07 |
| Pruned DDPM | 46.5M | 100.7G | 2.43GB | 4.13GB | 32.92 | 11.02 |

Table 3: The efficiency of pre-trained and pruned models

# 5 Training Details

## 5.1 Pruning pipeline

We follow a one-shot pipeline [3] to build an initial benchmark for diffusion model pruning:

- **Importance Estimation**: Importance estimation is performed in the group level [1]. We estimate the importance of weights directly on the pre-trained models, without any iterative strategies.
- **Pruning**: When unimportant parameters are identified, we physically remove those parameters to reduce model size. This is different from adding a mask to parameters, which only zeroes parameters.
- **Finetuning**: We follow the same training process as DDPMs [2]. After finetuning, we directly report the performance of the last checkpoint.

## 5.2 Hyper-parameters

The table summarizes the hyper-parameters employed in network pruning for diffusion models. It encompasses datasets such as CIFAR-10, CelebA-HQ, LSUN Church, and LSUN Bedroom, along with their respective image sizes. The hyper-parameters include pruning ratio (44%), learning rate (2e-4 or 2e-5), batch size (32, 96, or 128), and training steps (100K, 0.2M, or 0.5M). We set the weight decay to 0 for all datasets. These hyper-parameter values follows the training protocols of pre-trained models [2], except that we only finetune the pruned models for much less steps.

| Dataset | Img Size | Hyper-parameters | | | |
|---|---|---|---|---|---|
| | | Pruning Ratio | Lr | Batch | $\text{Step}_f/\text{Step}_p$ |
| CIFAR-10 | 32 | 44% (6.1G → 3.4G) | 2e-4 | 128 | 12.5% (100K/800K) |
| CelebA-HQ | 64 | 44% (23.9G → 13.3G) | 2e-4 | 96 | 20.0% (100K/500K) |
| LSUN Church | 256 | 44% (248.7G → 138.8G) | 2e-5 | 32 | 11.3% (0.5M/4.4M) |
| LSUN Bedroom | 256 | 44% (88G → 138.8G) | 2e-5 | 32 | 8.33% (0.2M/2.4M) |
| ImageNet | 256 | 47% (99.8G → 52.7G) | 2e-7 | 128 | 5.00% (0.1M/2M) |

Table 4: Hyper-parameters for our experiments. $\text{Step}_f$ and $\text{Step}_p$ refer to the number of steps for finetuning and pre-training respectively.

# 6 Visualization of Generated Images

In Figure 2 and 3, we visualize more images sampled from pre-trained models and pruned models. And in Figure 4, we also provide some failure cases with visual distortion or inconsistent contents. However, we find that our generated images still preserve similar contents to that from pre-trained models.

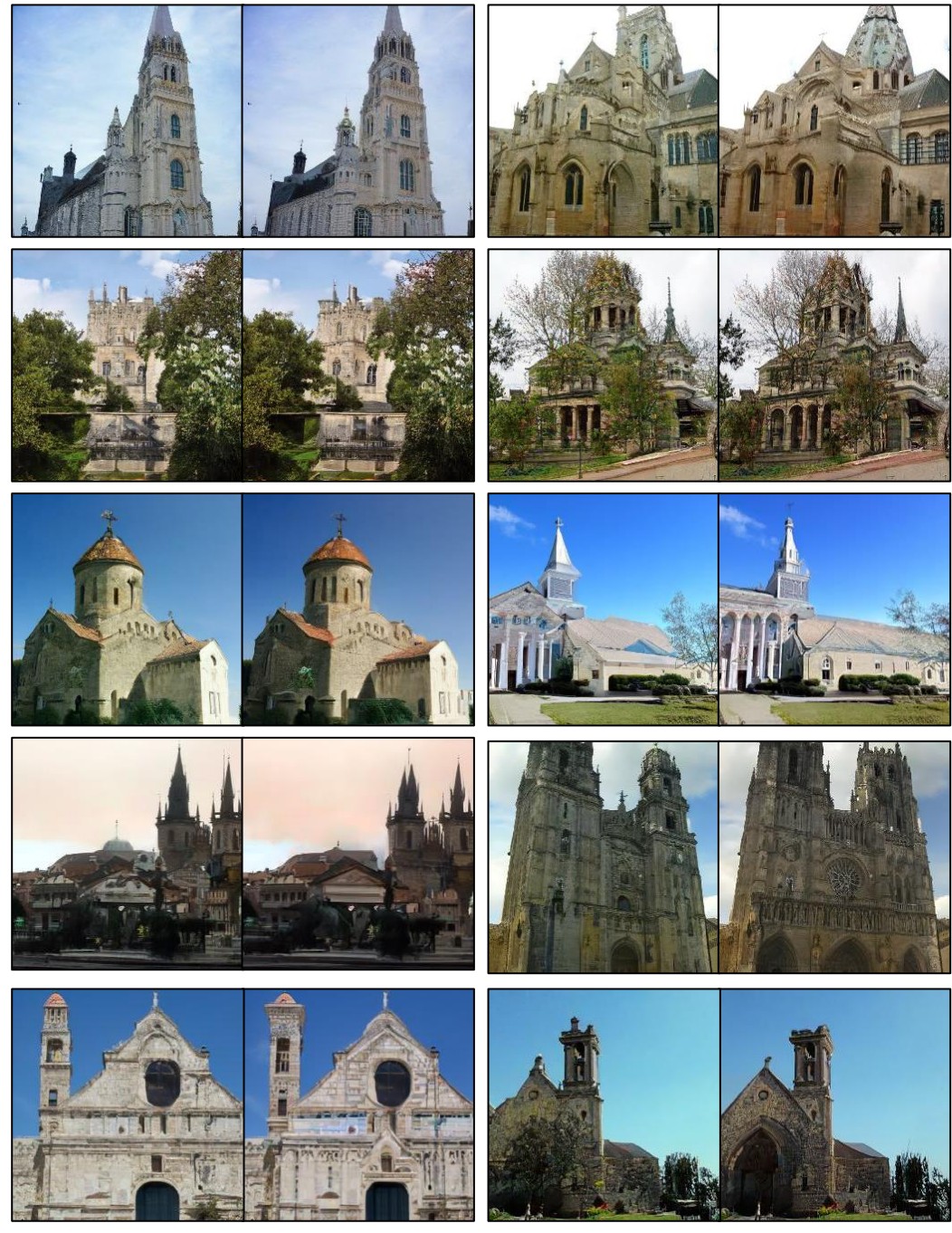

Figure 2: Church images sampled from pre-trained models (left) and pruned models (right).

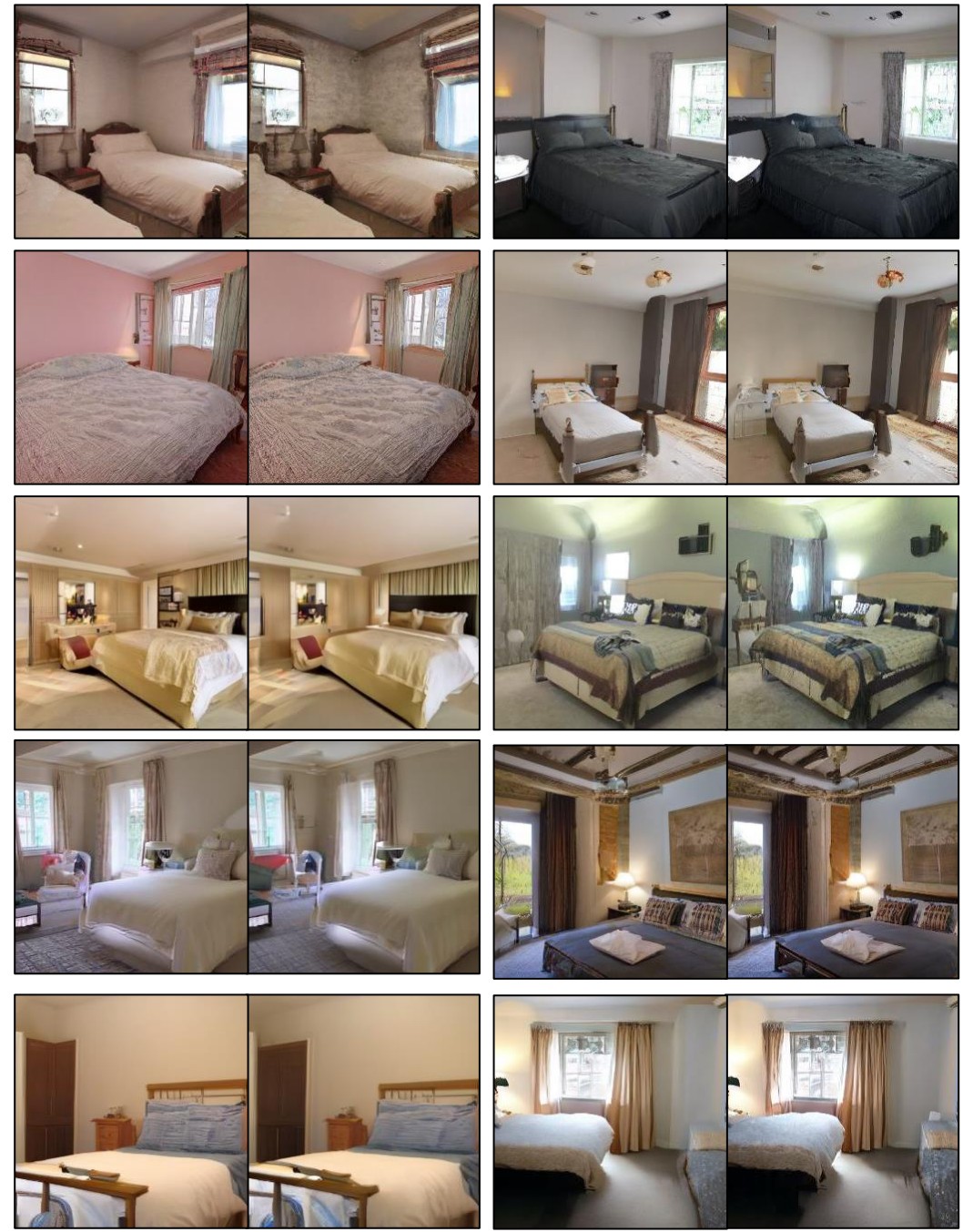

Figure 3: Bedroom images sampled from pre-trained models (left) and pruned models (right).

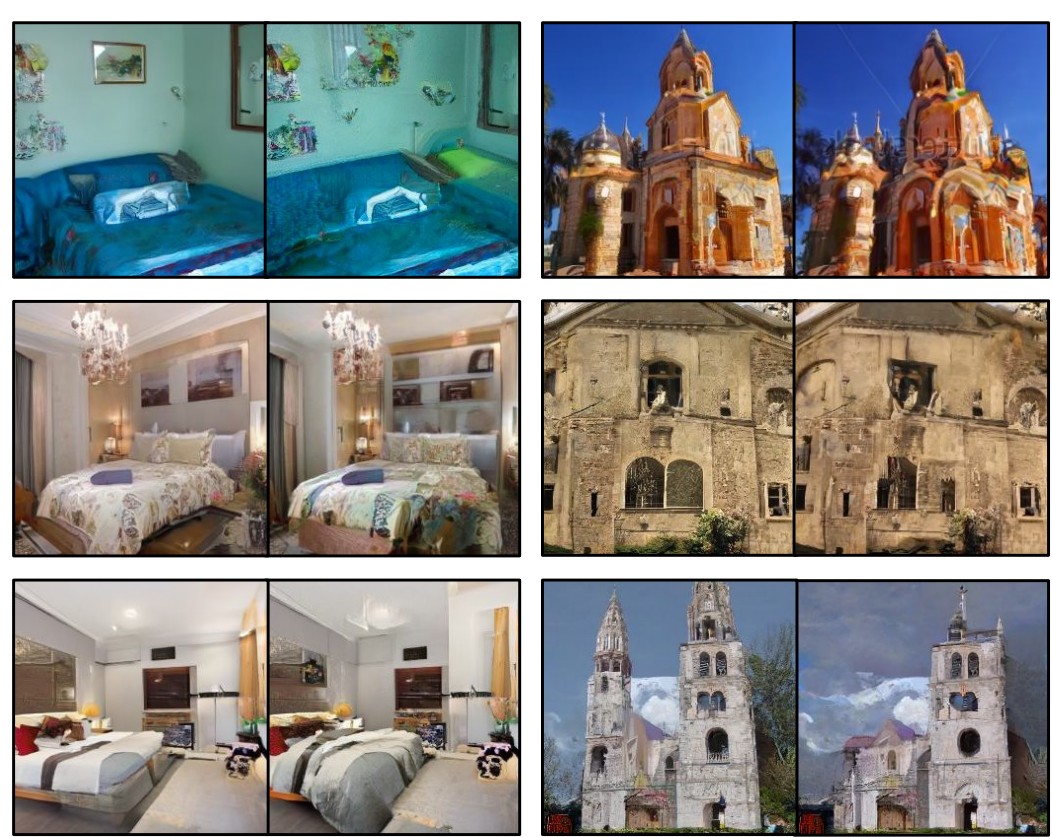

Figure 4: Failure cases sampled from pre-trained models (left) and pruned models (right).