# OpenReview forum: "Structural Pruning for Diffusion Models"
_NeurIPS.cc/2023/Conference — NeurIPS 2023 poster_

### Official Review · Reviewer_oE14 · 2023-06-28

**Soundness:** 3 good
**Presentation:** 3 good
**Contribution:** 3 good
**Rating:** 6
**Confidence:** 4

**Summary:**

This paper presents Diff-Pruning, an efficient compression method for learning lightweight diffusion models from pre-existing ones. DPMs have shown impressive capabilities in generative modeling, but they often come with significant computational overhead during training  and inference. Diff-Pruning addresses this challenge by introducing a  Taylor expansion over pruned timesteps, which eliminates  non-contributory diffusion steps and combines informative gradients to  identify important weights. The authors empirically evaluate the proposed method on four diverse datasets, highlighting two primary  benefits: 1) Efficiency, with a 50% reduction in FLOPs at a fraction of  the original training expenditure, and 2) Consistency, as the pruned  diffusion models retain generative behavior congruent with their pre-trained counterparts.


**Strengths:**

- The paper introduces a novel method, Diff-Pruning,  **specifically designed** for compressing diffusion models.
- The authors conduct empirical assessments on  four diverse datasets, providing a comprehensive analysis of the  proposed method's performance. The evaluation demonstrates the  effectiveness of Diff-Pruning in terms of efficiency and consistency.
- The paper appears to be well-written overall.


**Weaknesses:**

- Provide additional details about the pruning process: Although this paper describes the essence of diffo-pruning as Taylor expansions at pruning time steps, providing more specific details about the pruning process itself would enhance the clarity of the proposed method. Providing step-by-step explanations or pseudocode algorithms would help readers understand and replicate this approach.

- In Eq. (4), (5), (6), (10), (11), the symbols $|\cdot|$, $||\cdot||$, and $||\cdot||_0$ appear. Are they representing the same thing? The author needs to carefully check if the symbols in the formulas are correct. Additionally, the notation $\nabla L_t(\mathbf{\theta})(\mathbf{\theta}^\prime - \theta)$ should be reviewed for its validity since $\theta$ is a vector. Furthermore, what does the "$\cdot$" symbol represent in Equation (6)? Is it the dot product? If so, why is "$|\cdot|$" included?

- I am unsure how Eq. (6) is derived from Eq. (5). Why can it serve as an importance criterion? Does it have a direct relationship with the model's performance? If so, the author should demonstrate the relationship between this criterion and the performance of the diffusion model, such as through correlation analysis, and so on.

- Why does the pruned model exhibit better performance than the pretrained model? Can the author explain this phenomenon?

- I would like to know if the FID and SSIM metrics are sensitive to the pruned model.

- Can the MACs metric demonstrate the superiority of the proposed method in terms of performance acceleration? I am interested in understanding the effectiveness of the proposed method on training or inference speed under different GPU, e.g. Frame Per Second (FPS).

Overall, this article is well-written and easy to follow, but there are some questions that need to be addressed. If the author can address these questions, I would consider increasing the score.

**Questions:**

see Weaknesses.

**Limitations:**

The paper briefly mentions limitations but does not provide detailed explanations. Expanding on the limitations section will help readers understand the potential constraints or assumptions of the proposed method and its applicability in different scenarios. Additionally, it is necessary to discuss specific technical shortcomings.

---

> ### Author Rebuttal · Authors · 2023-08-09
>
> > **Q1:** Provide additional details about the pruning process like step-by-step explanations or pseudocode algorithms.
>
> **A1:** Thanks for the advice. Section 4 of the appendix provides a brief discussion about the pruning pipeline and training configurations. We will make it more detailed following your suggestions and provide pseudocode in the appendix or the main paper.
>
> ---
>
> > **Q2:** (a) The meaning of symbols $|\cdot|$, $||\cdot||$ and $||\cdot||_0$.
> > (b) Additionally, the notation  $\nabla L_t(\theta) (\theta^\prime - \theta)$ should be reviewed for its validity since  $\theta$ is a vector.
> > (c) Furthermore, what does the "$\cdot$" symbol represent in Equation (6)? Is it the dot product? If so, why is "$|\cdot|$" included?
>
> **A2:** Apologies for any confusion. We address the concerns as follows:
>
> * (a) In our formulation, $|\cdot|$ indicates the absolute value. The symbol $||\cdot||$, unless specified, corresponds to the L-2 Norm. Also, $||\cdot||$ denotes the L-0 norm, counting nonzero elements. To prevent confusion, we'll clarify this at the method's start. A typo in Line 118, $|\theta^\prime|_0$, will be corrected to $||\theta^\prime||_0$.
>
> * (b) In Eq. 5, $\nabla L_t(\theta) (\theta^\prime - \theta)$ is the dot product of vectors, yielding a scalar. For structural pruning, rows or columns of weight matrices are removed, warranting vector input for Taylor expansion.
>
> * (c) "$\cdot$" implies scalar multiplication. Eq. 6 gauges a weight's importance in the matrix. To estimate importance, we use the absolute value, as loss change can be negative (<0) or positive (>0), both impacting the model negatively.
>
> ---
>
> > **Q3:** (a) How Eq. (6) is derived from Eq. (5). Why can it serve as an importance criterion?
> > (b) Does it have a direct relationship with the model's performance? If so, the author should demonstrate the relationship.
>
> **A3:** The Taylor expansion in Eq. 2 measures the loss damage caused by pruning. By setting a single parameter scalar $ \theta_{ik} $ to 0 and keeping others unchanged. We will obtained a pruned weight vector $ (\theta^\prime_i - \theta_i) = [0, 0, ..., -\theta_{ik}, 0, 0, ..., 0] $, where all elements except $k$-th one are zero. If we apply the Taylor expansion in equation 2 and take the absolute value, we can obtain the importance for a parameter scalar:
>
> $\mathcal{I}(\theta_{ik}, x)= | \mathcal{L}_t(\theta^\prime) - \mathcal{L}_t(\theta) |$
>
> $ = | (\theta_{i0}-\theta_{i0}) \cdot \nabla_{\theta_{i0}}  + \dots + (0 - \theta_{ik}) \cdot \nabla_{\theta_{ik}} + \dots + (\theta_{iK}-\theta_{iK}) \cdot \nabla_{\theta_{iK}} | $
>
> $= |\theta_{ik} \cdot \nabla_{\theta_{ik}} |$
>
> For simplicity, we drop high-order term and use $\nabla_{\theta_{ik}}$ for $\nabla_{\theta_{ik}} L(\theta, x)$. If we set the whole vector  $\theta_i = [0, 0, ...]$, we get the "vector" criterion in Line 133: $| \sum_k \theta_{ik} \cdot \nabla_{\theta_{ik}} \mathcal{L}_t(\theta, x) |$. Thus, this criterion directly estimates performance damage: $| \mathcal{L}_t(\theta^\prime) - \mathcal{L}_t(\theta) |$. We show this relation in the PDF figure depicting generation quality from various pruning algorithms without tuning. Our algorithms outperform other methods in generation quality.
>
> ---
>
> > **Q4:** Why does the pruned model exhibit better performance than the pretrained model? Can the author explain this phenomenon?
>
> **A4**: This is akin to double descent [1], where pre-trained networks are oversized for certain datasets like CelebA, and pruning mitigates overfitting. CelebA images share similar features, allowing fewer parameters in the network. But this phenomenon was not observed in datasets like CIFAR-10, Church, and Bedrooms due to their varied content and complexity.
>
> [1] Sparse double descent: Where network pruning aggravates overfitting. ICML, 2022.
>
> ---
>
> > **Q5:** I would like to know if the FID and SSIM metrics are sensitive to the pruned model.
>
> **A5**: We provide an empirical study in Table 3 of the main paper. FID and SSIM (Structural Similarity) are both sensitive to the pruning ratio. With pruning increasing from 0% to 70%, FID climbs from 4.19 to 9.33, while SSIM drops from 1.00 to 0.909. So, FID and SSIM can indeed reveal the effectiveness of pruning.
>
> ---
>
> > **Q6**: Can the MACs metric demonstrate the superiority of the proposed method in terms of performance acceleration? I am interested in understanding the effectiveness of the proposed method on training or inference speed.
>
> **A6:** Thanks for the comments. Following the reviewer's advice, we provide more results in the Table below. Models are trained on 4090 but tested on A6000. MACs indeed reveals the actual speed-up.
>
> | Method    |  \#Params |  MACs | Inference Mem.  | Training Mem. | Inference FPS  | Training FPS  |
> |--|-----|----|----|--|--|--|
> | Pretrained LDM  | 400.92M  | 99.80G | 11.03GB | 14.64GB | 12.87 | 4.26 |
> | Pruned LDM  | 189.43M  | 52.71G | 9.58GB | 11.95GB| 19.83| 6.37 |
> | Pretrained DDPM | 113.7M| 248.7G| 3.35GB| 5.59GB| 28.66| 11.02|
> | Pruned DDPM| 46.5M| 100.7G| 2.43GB | 4.13GB | 32.92 | 9.07 |
>
> ---
>
> > **Q7:** The paper briefly mentions limitations but does not provide detailed explanations. Expanding on the limitations section will help readers understand the potential constraints or assumptions.
>
> **A7:** Thanks for the suggestion. We discussed some failure cases in Fig.4 of the appendix and in the experiments part of the main paper, such as Line 246 (about performance) and Line 253 (about distortion). We will provide a limitation section in the revised version for the following limitations and technical shortcomings:
>
> 1. Performance: it is still very difficult to preserve the original performance after pruning, especially on large-scale datasets.
> 2. Distortion: the algorithm only tries to minimize the global distortion, but is unaware of important semantic information in the model, such as watermarks.

---

> > ### Author Response · Authors · 2023-08-14
> >
> > We sincerely apologize for reversing the order of FPS for the pre-trained and pruned models in the table. The corrected version is as follows:
> >
> > | Method | #Params	|  MACs |	Inference Mem.	| Training Mem. |	Inference FPS | Training FPS |
> > | -- | -- | -- | -- | -- | -- | -- |
> > | Pretrained LDM	| 400.92M	| 99.80G	| 11.03GB	| 14.64GB	| 12.87	| 4.26|
> > | Pruned LDM	| 189.43M	| 52.71G	| 9.58GB	| 11.95GB	| 19.83	| 6.37|
> > | Pretrained DDPM	| 113.7M	| 248.7G	| 3.35GB	| 5.59GB	| 28.66	|  **9.07*** |
> > | Pruned DDPM	| 46.5M	| 100.7G	| 2.43GB	| 4.13GB	| 32.92	|  **11.02*** |
> >
> > Best Regards,
> > Authors of Paper 2537

---

> > ### Comment · Reviewer_oE14 · 2023-08-18
> >
> > Thank you for your detailed reply and I tend to raise my score.

---

> > > ### Author Response · Authors · 2023-08-18
> > >
> > > We extend our sincere gratitude to Reviewer oE14 for the valuable comments and suggestions. The points raised regarding symbol definitions, derivation details, limitations, FPS, and technical shortcomings are unquestionably important. In line with the reviewer's insightful feedback, we will incorporate the above results and analyses into the revised version.
> > >
> > > Best Regards,
> > > Authors of Submission 2537

---

### Official Review · Reviewer_5jdL · 2023-07-04

**Soundness:** 2 fair
**Presentation:** 2 fair
**Contribution:** 2 fair
**Rating:** 5
**Confidence:** 4

**Summary:**

This work introduces a structural pruning method for diffusion models, called Diff-pruning. The authors leverage Taylor expansion on each term of the ELBO loss as the creteria to decide the pruned weights. By calculating the error of the final image induced by the pruning at timestep t and discussing about the effect of converged loss on the higher order terms of the Taylor expansion, this work manually drop the pruning creteria of timesteps near noise by setting a threshold $\mathcal{T}$ related to the relative loss. Experiments compare the performance of the proposed pruning method with other common pruning methods, as well as training from scratch.

**Strengths:**

1. This paper proposes a novel pruning method that utilizes the multi-step property of the diffusion model.
2. The experiment results demonstrate the effectiveness of the proposed method compared with baseline methods.

**Weaknesses:**

1. My major concern is the motivation of the method is not clearly explained:
  - The authors conclude an important conclusion from eq.9. However, the derivation process of eq.9 has mistakes and typos. There is a neural inference term with $x_t$ as input in eq.3, the final error $\delta_0$ can not be derived in such a simple form. Moreover, the meaning of the symbol "$\delta$" is ambiguous. It seems to represent noise prediction error in line 146, but $\delta_{t-1}$ and $\delta_0$ seems to represent the error of $x_t$ and $x_0$ caused by pruning. And the subscript $s$ of $\alpha_s$ is wrongly written in eq.8.
  - How could Equ. 9 derive the conclusion "prediction errors occurring at larger t primarily impact the high-level content of generated images, while smaller t values concentrate on refining the images with relatively small modifications"? Relevant paper presents that the main difference between high-level contents and details is that they come from different frequency components of the whole image [1] , while Equ.9 focus on the "amplitude of the error $\delta_0$". It requires further clarification.
  - The authors claim that the final distortion is progressively magnified according to eq.9, which indicates that the error at timesteps near the noise has a greater impact on the final image. However, the proposed method truncates the pruning criteria at large t, which does not match such observation.

2. Lack of detailed description of the baseline method ToMe.

[1] Diffusion probabilistic model made slim. arXiv preprint arXiv:2211.17106, 2022.

**Questions:**

1. How is eq.9 derived and why the choice of $\alpha_t$ according to this equation is opposite to the proposed method (as described in Weaknesses)?

**Limitations:**

The authors do not include the limitations and potential negative societal impact in the paper.

---

> ### Author Rebuttal · Authors · 2023-08-09
>
> > **Q1:** The motivation of the method is not clearly explained.
>
> **A1:** In essence, we harness the informative gradients stemming from early timesteps ($t\rightarrow 0$) to identify unimportant parameters for the pruning process. As demonstrated in our experiments, we disregard larger timesteps ($t\rightarrow T$) due to the presence of vanished gradients, rendering them incapable of furnishing valuable insights for importance assessment.
>
> ---
>
> > **Q2:** The derivation process of eq.9 has mistakes and typos. (a) There is a neural inference term with $x_t$ as input in eq.3, the final error $\delta_0$ can not be derived in such a simple form. (b) Moreover, the meaning of the symbol "$\delta$" is ambiguous. It seems to represent noise prediction error in line 146, but $\delta_{t-1}$ and $\delta_0$ seems to represent the error of $x_t$ and $x_0$ caused by pruning. (3) And the subscript $s$ of $\alpha_s$ is wrongly written in eq.8.
>
> **A2:** I apologize for any confusion that may have arisen. In Equation 9, $\delta_t=\epsilon_{\theta^\prime }(x, t) - \epsilon_{\theta}({x}, t)$ (as defined in Line 146) denotes the error that emerges under the same input. On the other hand, $\delta_0$ refers to the distortion resulting from the pruning error $\delta_t$, which **does not include the error associated with input shift** due to the difficulty in analyzing the non-linearity of network forwarding. In order to enhance clarity, we would like to improve the notation by substituting $\delta_0$ with $\delta_{0\leftarrow t}$. The subsequent discussion employs this refined notation for clarification.
>
> * (a) Indeed, there's a network inference $\epsilon_\theta (x_{t-1} + \delta_{t-1}, t-1)$ in different timessteps. At Line 147, we assume "no additional prediction error emerges in other steps," indicating consistent inference across timesteps without introducing new pruning errors. Besides, we don't concentrate on changes caused by shifted inputs as they can't reflect the **functional shifts** due to pruning. Instead, we are interested in the pruning error as defined above, that is $\delta_{t-1}=\epsilon_{\theta'}(x_{t-1} + \delta_{t-1}, t-1) - \epsilon_{\theta}(x_{t-1} + \delta_{t-1}, t-1)$ . In this case, a simple functional error can be linearly derived as $\delta_{0\leftarrow t}$ in Eq. 9. However, there are still error terms caused by the input shift during inference, i.e., $\epsilon_{\theta}(x_{t-1} + \delta_{t-1}, t-1) - \epsilon_{\theta}(x_{t-1}, t-1)$. Analyzing this is challenging due to non-linearity. We'll revise Eq. 9 per your suggestion to address these concerns.
>
> * (b) Thanks for the comments. As defined in Line 146, $\delta_t=\epsilon_{\theta^\prime}(x, t) - \epsilon_\theta (x, t)$. It refers to the functional error with the same inputs.
>
> * (c) Thanks. The $\alpha_s$ should be $\alpha_t$. We will make the necessary correction to fix it.
>
> ---
>
> > **Q3:** How could Equ. 9 derive the conclusion "prediction errors occurring at larger t primarily impact the high-level content of generated images, while smaller t values concentrate on refining the images with relatively small modifications"?.
>
> **A3:** Thanks for the comment. Due to the scaling factor mentioned in Line 149, "the final distortion induced by $\delta_t$ is progressively magnified by a factor of $\frac{1}{\sqrt{\alpha_s}}>1$ along the sampling path". For a larger $t$, more scaling factors will be applied. Therefore the distortion at a large $t$ has a larger impact ($\delta_{0\leftarrow t}$) on the image space than that at a small $t$.
>
> ---
>
> > **Q4:** The authors claim that the final distortion is progressively magnified according to eq.9, which indicates that the error at timesteps near the noise has a greater impact on the final image. However, the proposed method truncates the pruning criteria at large t, which does not match such observation.
>
> **A4:** Thanks for the comment. The impact have two main components: the scaling factor $\frac{\beta_t}{\sqrt{\bar{\alpha}_t \cdot (1-\bar{\alpha}_t) } }$ and the initial error $\epsilon_t$ at step $t$. While it might seem reasonable to prioritize focusing on larger steps $t$ as a potential solution, there are certain challenges associated with this approach. As detailed in Line 172-177 and depicted in Figure 4, the gradients tend to vanish rapidly as $t$ approaches $T$, leading to inaccuracies in the applied Taylor expansion for $\delta_t$. Consequently, it becomes essential to truncate certain steps to prevent the accumulation of unreliable gradients.
>
> ---
>
> > **Q5:** Lack of detailed description of the baseline method To Me.
>
> **A5:** Thank you for your valuable feedback. The specific details of the baseline methods can be found in Lines 218-228 of the main paper. We will follow your suggestion to make it an independent paragraph in the revised version.

---

> > ### Comment · Reviewer_5jdL · 2023-08-18
> > **Thanks for the authors' response**
> >
> > The authors' response addresses most of my concerns. Therefore, I'm increasing my score.
> >
> > BTW, I still have several questions.
> > * A2 (a): According to my understanding, eq 9 assumes that the shift of $x_{t-1}$ (caused by pruning) does not affect functional shift at timesteps smaller than $t$, so that this error can be passed to $x_0$ in the way of sequenced multiplication. However, since neural network is a very complicated non-linear term that may have a significant effect on the final error. So, will this assumption be too strong to be true? What're the authors' opinions?
> > * A3: My major concern is that the "high-level" details and "low-level" contents are directly related to the **frequency** of the image, not the amplitude of the error. So maybe it's inappropriate to derive the conclusion at Line 150-152 from eq 9.

---

> > > ### Author Response · Authors · 2023-08-18
> > >
> > > We sincerely thank Reviewer 5jdL for the valuable comments and suggestions to improve our submission. And we will follow the reviewer's advice to polish our submission.
> > >
> > > > However, since neural network is a very complicated non-linear term that may have a significant effect on the final error. So, will this assumption be too strong to be true? What're the authors' opinions?
> > >
> > > We agree with the reviewer that the neural network is complicated due to the nonlinearity, which indeed has a significant effect on the final error. Equation 9 only provides a coarse estimation of the partial effect of pruning, under a ideal and strong assumption. So, we only use it as an interpretation of why we need to pay more attention to large timesteps. This is validated in Fig. 4 where incorporating some large timesteps can be beneficial to the image quality (SSIM, 0.78 $\rightarrow$ 0.82). We will polish Lines 141-149 following the reviewer's comment to emphasize that Eq.9 is a coarse estimation under strong assumptions and make the presentation more rigorous.
> > >
> > > > My major concern is that the "high-level" details and "low-level" contents are directly related to the frequency of the image, not the amplitude of the error. So maybe it's inappropriate to derive the conclusion at Line 150-152 from eq 9.
> > >
> > > We appreciate the comments. This conclusion is inspired by Figure 6 in DDPM, where they find that "Large scale image features
> > > appear first and details appear last."[1]. However, we agree with the reviewer that frequency is a more essential factor for this phenomenon. But to some extent, the magnitude of the error can also reveal the type of distortion as content changes inherently lead to substantial errors in the pixel space. Following the reviewer's comment, we will provide a discussion about the frequency and the error in the revised version.
> > >
> > > [1], Ho, Jonathan, Ajay Jain, and Pieter Abbeel. "Denoising diffusion probabilistic models." Advances in neural information processing systems 33 (2020): 6840-6851.
> > >
> > > Best Regards,
> > > Authors of Submission 2537

---

### Official Review · Reviewer_i6fA · 2023-07-05

**Soundness:** 4 excellent
**Presentation:** 3 good
**Contribution:** 4 excellent
**Rating:** 7
**Confidence:** 4

**Summary:**

The paper introduces a pruning-based method aimed at addressing the efficiency challenges of diffusion models. Unlike existing approaches that focus on accelerating sampling or enhancing architectures, it specifically focus on the time cost of compression and the consistency in generated images. The authors propose a novel variant of Taylor approximation as the importance function for pruning, which effectively preserves both high-level content and low-level details of generated images, while minimizing the impact of noisy steps. Experimental results demonstrate that the compressed diffusion model successfully generates similar images to the pre-trained model.

**Strengths:**

I agree that efficiency and consistency are indeed crucial considerations in compressing diffusion models, which have received limited attention in prior works. This work contributes to this field by building several initial experiments on datasets such as Cifar, CelebA, and LSUN, effectively showcasing the advantages of the method. This work establishes a solid baseline for future explorations into efficient diffusion models. Moreover, the impressive ability of the pruned model to generate consistent results is particularly noteworthy, which is user-friendly in deployment.

**Weaknesses:**

It is observed that in certain cases, the generated images may still exhibit some distortions or changes, such as the watermarks shown in Figure 2. It is worth noting that some of these elements may contain important information that should ideally be preserved. To address this, it would be beneficial to explore methods that allow for controllable preservation of such information while pruning the models. This ability would provide the flexibility to focus on specific parts of the generated images based on different scenarios.

**Questions:**

It would be valuable for the author to include discussions regarding the controllability of the observed distortions or changes, as mentioned in the weaknesses.

---

> ### Author Rebuttal · Authors · 2023-08-09
>
> > **Q1:** the generated images may still exhibit some distortions or changes, such as the watermarks shown in Figure 2. A discussion about controllable preservation of such information. it would be beneficial to explore methods that allow for controllable preservation of such information while pruning the models. This ability would provide the flexibility to focus on specific parts of the generated images based on different scenarios.
>
> **A1:** Thanks for your valuable comments. This work focuses on minimizing distortion at the pixel level and does not inherently offer direct controllability at a high level. Nevertheless, the idea of incorporating such functionality is highly valuable. To achieve this, one solution is to extend our method by utilizing a mask that prioritizes important regions, employing the following equation adapted from Equation 2 in the main paper:
>
> $$
> \mathcal{L}( {\theta} ):= \mathbb{E}_{t, x_0\sim q(x), \epsilon\sim \mathcal{N}(0,1)} [ || m \odot ( \epsilon  -  \epsilon^{\prime}(\sqrt{\bar{\alpha}_t}x_0 + \sqrt{1-\bar{\alpha}_t}\epsilon, t) ) ||^2 ]
> $$
>
> where $m$ is a binary mask in the image space and $\odot$ is an element-wise multiplication. Here we use $\epsilon^{\prime}$ instead of "\epsilon_{\theta}" because there is a display bug in OpenReview. This method allows the pruner to perverse those highlighted regions like the watermarks. However, it is important to note that this improvement relies on the availability of additional annotations. To address this limitation, we can explore the possibility of decoupling objects in images and achieving a balanced quality for each object. A naive way is to segment the images with general models like SAM [1]. We intend to include a discussion section in the appendix that delves into the controllability of image quality and its implications.
>
> [1] Kirillov, Alexander, et al. "Segment anything." arXiv preprint arXiv:2304.02643 (2023).

---

### Official Review · Reviewer_PXRJ · 2023-07-07

**Soundness:** 4 excellent
**Presentation:** 4 excellent
**Contribution:** 4 excellent
**Rating:** 5
**Confidence:** 3

**Summary:**

This paper presents Diff-Pruning, a novel structural compression technique for learning efficient diffusion models from pre-trained ones. The fundamental idea behind Diff-Pruning is the utilization of Taylor expansion on pruned timesteps, which effectively combines informative and clean gradients to estimate the importance of weights. The authors demonstrate that Diff-Pruning not only achieves compression of pre-trained models within a few training iterations but also maintains the original generation capabilities intact.

**Strengths:**

- The experimental results demonstrate that structural pruning can serve as a powerful and efficient compressor for diffusion models. An important advantage is that the pruned model inherently retains the generative behavior of the pre-trained model. Considering the time-consuming nature of training new diffusion models, this work proves valuable for various downstream applications.
- The proposed method, which employs the Taylor expansion over pruned timesteps, is both well-motivated and practical. The idea of balancing the contribution of different stages with binary weighting is interesting and easy to practice.
- The authors provide extensive experimental evidence that substantiates their claims regarding the efficiency and consistency of Diff-Pruning.

**Weaknesses:**

- Extra comparison of the memory requirement for different pruning methods might provide a more complete assessment of the proposed methods.
- The experiments reveal that structural pruning can potentially have a negative impact on the performance of pre-trained models. This observation deviates slightly from the results seen in discriminative tasks such as classification, where lossless compression can be achieved for certain networks like ResNets. It would be beneficial for the authors to provide further clarification on this phenomenon

**Questions:**

Please refer to the weaknesses above.

**Limitations:**

This work has no negative societal impact.

---

> ### Author Rebuttal · Authors · 2023-08-09
>
>
>
> > **Q1:** Extra comparison of the memory requirement.
>
> **A1:** Thanks for the suggestion. Methods like Random Pruning and Magnitude Pruning do not require additional memory during pruning. Taylor pruning and the proposed diff-pruning require $O(N)$ space to store the gradient, where $N$ is the number of parameters. In addition, the memory consumption of pruned models, as well as their training and inference FPS (Frames Per Seconds), can be found in the table below. We tested a pruned DDPM and a conditional LDM (Please refer to the Q1 of Reviewer S8k8) on 256$\times$256 images, i.e., LSUN-Church and ImageNet-1K, with a single NVIDIA RTX A5000. All experiments were repeated for 30 times and the average results are reported.
>
> | Method             |  \#Params |  MACs | Inference Mem.  | Training Mem. | Inference FPS  | Training FPS  |
> |------------------------|---------------------------|-----------------------|--------------------------------|-----------------------------|------------------------------|--------------------------|
> | Pretrained LDM         | 400.92M                   | 99.80G                | 11.03GB                        | 14.64GB                     | 12.87                        | 4.26                     |
> | Pruned LDM             | 189.43M                   | 52.71G                | 9.58GB                         | 11.95GB                     | 19.83                        | 6.37                     |
> | Pretrained DDPM        | 113.7M                    | 248.7G                | 3.35GB                         | 5.59GB                      | 28.66                        | 11.02                    |
> | Pruned DDPM            | 46.5M                     | 100.7G                | 2.43GB                         | 4.13GB                      | 32.92                        | 9.07                     |
>
> ---
>
> > **Q2:** The negative impact on the performance of pre-trained models. Why Lossless compression can not be achieved.
>
> **A2:** Thank you for your insightful comments. We propose two factors that may cause this phenomenon:
>
> * Model Capacity: The model capacity required for generative models is typically larger compared to discriminative models [1,2,3]. In discriminative tasks, images are often downsampled and filtered hierarchically, resulting in less information encoded within the network. Conversely, generative tasks face challenges in training small networks while achieving high generation quality, which causes performance lost during pruning.
>
> * Sensitivity of Metrics: Classification metrics like accuracy are generally not very sensitive to slight distortion in model predictions, as long as the samples maintain their correct positioning in relation to the decision boundary. In other words, Accuracy is discrete. However, for diffusion models, we use FID, a continuously changed value, to evaluate the performance of models. Pruning will introduce distortions in the generated images, which are immediately reflected in the FID score.
>
> [1] Kang, Minguk, et al. "Scaling up gans for text-to-image synthesis." Proceedings of the IEEE/CVF Conference on Computer Vision and Pattern Recognition. 2023.
> [2] Brock, Andrew, Jeff Donahue, and Karen Simonyan. "Large scale GAN training for high fidelity natural image synthesis." arXiv preprint arXiv:1809.11096 (2018).
> [3] Rombach, Robin, et al. "High-resolution image synthesis with latent diffusion models." Proceedings of the IEEE/CVF conference on computer vision and pattern recognition. 2022.

---

> > ### Author Response · Authors · 2023-08-14
> >
> > We sincerely apologize for reversing the order of FPS for the pre-trained and pruned models in the table. The corrected version is as follows:
> >
> > | Method | #Params	|  MACs |	Inference Mem.	| Training Mem. |	Inference FPS | Training FPS |
> > | -- | -- | -- | -- | -- | -- | -- |
> > | Pretrained LDM	| 400.92M	| 99.80G	| 11.03GB	| 14.64GB	| 12.87	| 4.26|
> > | Pruned LDM	| 189.43M	| 52.71G	| 9.58GB	| 11.95GB	| 19.83	| 6.37|
> > | Pretrained DDPM	| 113.7M	| 248.7G	| 3.35GB	| 5.59GB	| 28.66	|  **9.07*** |
> > | Pruned DDPM	| 46.5M	| 100.7G	| 2.43GB	| 4.13GB	| 32.92	|  **11.02*** |
> >
> > Best Regards,
> > Authors of Paper 2537

---

### Official Review · Reviewer_S8k8 · 2023-07-11

**Soundness:** 2 fair
**Presentation:** 2 fair
**Contribution:** 2 fair
**Rating:** 4
**Confidence:** 2

**Summary:**

The paper proposed a method to prune diffusion models to achieve 50% FLOPS reduction at 10% to 20% of original training budget. The authors show that prune and fine-tune strategy based on the random selection, magnitude based selection, Taylor expansion based selection all lead to non-ideal / suboptimal performance. They develops a method based on the modification of Taylor expansion based selection and show it achieves good performance-quality tradeoff. They quantitatively evaluated on standard datasets used for diffusion models at small and medium resolution (from 32^2 to 256^2).

**Strengths:**

The paper works on a popular task in recent literature of generative models and presented a practice to prune diffusion models while preserves its generative capability. They demonstrated empirical results on 256^2 resolution unconditional image generation, which seems promising.

**Weaknesses:**

The lack of objective quantitative metrics in image generation literature makes the evaluation of pruned model very difficult. In addition to FID score, the authors also use SSIM score to compare the image generated from the full model and the pruned model under the same random seed is used. However, this is still hard for me to evaluate the quality of empirical results given limited examples presented in the paper. It would be otherwise more informative, if the authors can demonstrate their method being valid for conditional diffusion models, or even latent diffusion models.


The technical contribution is also limited, as the method authors used already exist in the literature but has not been adopted for diffusion models.



**Questions:**

I don't have other questions

**Limitations:**

I have a feeling that authors tried very hard to prove their technique working to some extent. It would be helpful as a research publication if the authors can present more results telling when and where their method fails or explore more on the boundary of quality and performance trade-offs.

---

> ### Author Rebuttal · Authors · 2023-08-09
>
> > **Q1:** The lack of objective quantitative metrics, and the effectiveness on conditional diffusion models like LDMs.
>
> **A1:** Thanks for your valuable comments. In the table below, we provide the results of pruning Conditional Latent Diffusion Models on ImageNet-1K (256$\times$ 256). We follow the same training protocol as provided in \cite{rombach2022high} to prune and fine-tune an off-the-shelf LDM. Due to the time limits of this rebuttal period, we only fine-tuned the pruned models for 4 epochs. The generated images can be found in the attached pdf file.
>
> |  Method         |  #Params $\downarrow$ | MACs $\downarrow$   |  FID $\downarrow$ |  IS $\uparrow$ |  Train Steps $\downarrow$ |
> |--------------------------|---------------------------|-------------------------|----------------------|-------------------|------------------------------|
> | Pretrained LDM           | 400.92M                   | 99.80G                  | 3.60                 | 247.67            | 2000K                        |
> | Scratch Training         | 189.43M  | 52.71G | 51.45                | 25.69             | 100K                         |
> | Taylor Pruning           |       189.43M                      |       52.71G                   | 11.18                | 138.97            | 100K                         |
> | Ours ($\mathcal{T}=0.1$) |         189.43M                    |         52.71G                 | 9.16                 | 201.81            | 100K                         |
>
> **Details of LDM Pruning**
> Here we also provide the details of LDM pruning: LDM consists of an encoder, a decoder, and a UNet diffusion model. Around 400M parameters come from the UNet architecture and only 55M from the autoencoder. So we only prune the UNet model for acceleration. Moreover, we are dealing with a **conditional** model. During importance estimation, we randomly sample classes and images to accumulate gradients for Taylor expansion. We used the threshold $\mathcal{T}=0.1$ to ignore those vanished gradients from large timesteps, which also makes the pruning process more efficient. With $T=0.1$, only 534 steps participate in the pruning process. After importance estimation, we apply a pre-defined channel sparsity of 30\% to all layers, leading to a lightweight UNet with 189.43M parameters. Subsequent fine-tuning follows official training scripts, using a scaled learning rate of $0.1\times lr_{base}$.  After fine-tuning, we sample 50 images per class and report the FID and IS score of the pruned models. Besides, We only report the #Params and MACs of the UNet. Similar to the hyper-parameter tables in the appendix, we outline our training configurations as follows:
>
> |  Dataset | Img Size |  Pruning Ratio (MACs)    |  Lr  |  Batch |  $\text{Step}_f/\text{Step}_p$ |  weight decay |
> |-------------|----------|--------------------------------------|---------|-----------|-----------------------------------|--------|
> |    ImageNet-1K    | 256      | 47.2\% (99.80G $\rightarrow$ 52.71G) | 1.28e-5 | 64        | 5\% (100K/2000K)                  | 0      |
>
> ---
>
> > **Q2:** The technical contribution is also limited, as the method authors used already exists in the literature but has not been adopted for diffusion models.
>
> A2: Thanks for the comments. Pruning diffusion models is quite a new topic and there is a large design space to explore, such as importance estimation, fine-tuning, and pruning strategies. This work indeed adopts some popular methods like Magnitude Pruning and Taylor Pruning to diffusion models.  However, as illustrated in Table 1, they do not show significant improvements even compared to random pruning. As an initial exploration, we aim to make the method as simple and practical as possible. This also brings the benefit that many existing techniques like second-order approximation can be further leveraged to boost the performance of pruning. We will follow your suggestion to make it a more "diffusion-style" work.
>
> ---
>
> > **Q3:** It would be helpful as a research publication if the authors can present more results telling when and where their method fails or explore more on the boundary of quality and performance trade-offs.
>
> **A3:** Thanks for the comment. We provided some failure cases in Fig. 4 of the appendix and also discussed some limitations in the controllability, such as the missing watermark in Line 252. In other words, diff-pruning only aims to minimize image distortion but does not know what kind of information is important in the images. We will provide a limitation section in the revised version. And for the boundary of quality and performance trade-offs, Table 3 provides some insights into the trade-off between performance and efficiency, where we observed prominent performance degradation with high pruning ratios. However, the pruning of diffusion models as well as generative models is till a new topic. I agree that more explorations on the boundary of quality and performance will be valuable.

---

### Author Rebuttal · Authors · 2023-08-09

We thank all reviewers for their constructive comments. Following the advice of Reviewer S8k8, we report our results on conditional LDMs on ImageNet-1K 256. Generated images are available in the attached PDF file.

|  Method         |  #Params $\downarrow$ | MACs $\downarrow$   |  FID $\downarrow$ |  IS $\uparrow$ |  Train Steps $\downarrow$ |
|--------------------------|---------------------------|-------------------------|----------------------|-------------------|------------------------------|
| Pretrained LDM           | 400.92M                   | 99.80G                  | 3.60                 | 247.67            | 2000K                        |
| Scratch Training         | 189.43M  | 52.71G | 51.45                | 25.69             | 100K                         |
| Taylor Pruning           |       189.43M                      |       52.71G                   | 11.18                | 138.97            | 100K                         |
| Ours ($\mathcal{T}=0.1$) |         189.43M                    |         52.71G                 | 9.16                 | 201.81            | 100K                         |

|  Dataset | Img Size |  Pruning Ratio (MACs)    |  Lr  |  Batch |  $\text{Step}_f/\text{Step}_p$ |  weight decay |
|-------------|----------|--------------------------------------|---------|-----------|-----------------------------------|--------|
|    ImageNet-1K    | 256      | 47.2\% (99.80G $\rightarrow$ 52.71G) | 1.28e-5 | 64        | 5\% (100K/2000K)                  | 0      |

---

> ### Comment · Area_Chair_dc8K · 2023-08-18
>
> Dear Authors,
>
> Thank you for your thorough and detailed responses to the reviewers' comments. We appreciate the efforts you have made to address each point raised by the reviewers and provide insightful explanations.
>
> Your responses have been carefully reviewed, and we will take them into full consideration during our discussions.
>
> Best regards,
>
> Area Chair

---

### Decision · Program_Chairs · 2023-09-21

**Decision:**

Accept (poster)

**Comment:**

The paper introduces Diff-Pruning, a novel structural compression technique for learning efficient diffusion models from pre-trained ones. The method utilizes Taylor expansion on pruned timesteps to estimate the importance of weights, achieving a good performance-quality tradeoff. Diff-Pruning successfully compresses pre-trained models within a few training iterations while maintaining the original generation capabilities intact.

The reviewers raised several concerns regarding the evaluation metrics, the technical contribution, the impact of structural pruning on performance, the preservation of important information in generated images, the derivation and clarification of equations, the lack of detailed baseline description, and the relationship between the proposed criterion and model performance.

After rebuttal, the authors have addressed most of the concerns. As a result, the paper received a score of 4-5-5-6-7, with the majority of reviewers agreeing to accept it. Considering the reviewer who gave a score of 4 expressed lower confidence (2), the AC has agreed to accept the paper and advises the authors to consider the reviewers' feedback and incorporate the rebuttal content into the final version.